# Do data-driven models beat numerical models in forecasting weather extremes? A comparison of IFS HRES, Pangu-Weather and GraphCast

Leonardo Olivetti[1,2,3] and Gabriele Messori[1,2,4]

[1]Department of Earth Sciences, Uppsala University, 75236 Uppsala, Sweden
[2]Swedish Centre for Impacts of Climate Extremes (climes), Uppsala University, 75236 Uppsala, Sweden
[3]Centre of Natural Hazards and Disaster Science (CNDS), Uppsala University, 75236 Uppsala, Sweden
[4]Department of Meteorology and Bolin Centre for Climate Research, Stockholm University, 10691 Stockholm, Sweden

**Correspondence:** Leonardo Olivetti (leonardo.olivetti@geo.uu.se)

**Abstract.** The last few years have witnessed the emergence of data-driven weather forecast models capable of competing and in some respects outperforming physics-based numerical models. However, recent studies question the capability of data-driven models to provide reliable forecasts of extreme events. Here, we aim to evaluate this claim by comparing the performance of leading data-driven models in a semi-operational setting, focusing on the prediction of near-surface temperature and windspeed extremes globally. We find that data-driven models mostly outperform ECMWF's physics-based deterministic model in terms of global RMSE for 1d-10d ahead forecasts, and can also compete in terms of extreme weather predictions in most regions. However, the performance of data-driven models varies by region, type of extreme event, and forecast lead time. Notably, data-driven models appear to perform best for temperature extremes in regions closer to the tropics and at shorter lead times. We conclude that data-driven models may already now be a useful complement to physics-based forecasts in regions where they display superior tail performance, but that some challenges still need to be overcome prior to operational implementation.

## 1 Introduction

The first deep learning models for weather applications date back to the 1990s (Schizas et al., 1991; Hall et al., 1999), but it is only in recent years that deep learning models have become competitive as self-standing medium-range forecasting tools. Since 2022, at least eight different research groups (Pathak et al., 2022; Bi et al., 2023; Keisler, 2022; Lam et al., 2023; Chen et al., 2023a; Nguyen et al., 2023; Chen et al., 2023b; Lang et al., 2024) claim to have developed deep learning models able to produce more accurate deterministic forecasts than state-of-the-art physics-based models from the European Centre for Medium-Range Weather Forecasts (ECMWF) in a range of atmospheric variables at multiple lead times. Recent independent studies (Rasp et al., 2024; Bouallègue et al., 2024) support these claims, showing how the data-driven models can outperform physics-based models in a wide range of parameters and metrics. In particular, the WeatherBench 2 (Rasp et al., 2024) provides comprehensive global and regional scorecards for comparing forecast models in terms of test-sample RMSE, while also making all test predictions produced freely available to the public.

However, the studies conducted so far focus on the average skill of the forecasts, without any special treatment of extreme events. Even if some cases studies have been conducted, for instance on cyclone tracking (Charlton-Perez et al., 2024; Bi et al., 2023; Lam et al., 2023; Chen et al., 2023b) and surface temperature extremes (Bouallègue et al., 2024; Lam et al., 2023), these are too limited to allow for a fair assessment of the capacity of data-driven models to forecast weather extremes globally. Timely and reliable forecasting of weather extremes plays a key role in disaster management and risk mitigation (World Meteorological Organization, 2022; Merz et al., 2020) and in crucial socio-economic functions, such as the energy and insurance sectors (e.g. Kron et al., 2019). We thus argue that greater emphasis should be placed on understanding whether data-driven models can provide reliable forecasts of weather extremes before such models may be implemented operationally (Watson, 2022).

In addition, recent studies (Watson, 2022; Olivetti and Messori, 2024; de Burgh-Day and Leeuwenburg, 2023) problematise the assumption that a strong performance in standard metrics of average skill should translate by default into an equally strong performance in the tails of the distribution. Indeed, there may be several reasons for an asymmetry between average skill and skill for extremes, including the intrinsic sparsity of extreme events in training datasets (Watson, 2022), the use of symmetric loss functions that are inadequate for extremes (Xu et al., 2024; Olivetti and Messori, 2024), and the multitask and multi-step optimisation approaches used in leading deep learning architectures (e.g. Bi et al., 2023; Lam et al., 2023). These issues are further exacerbated by the fact that the current generation of data-driven models published in peer-reviewed journals provides deterministic predictions, even though a number of promising approaches to provide uncertainty estimates for the predictions exist for older data-driven models (e.g. Scher and Messori, 2021; Clare et al., 2021) and are currently being explored for state-of-the-art models (e.g. Price et al., 2024; Hu et al., 2023; Bi et al., 2023; Zhang et al., 2023; Cisneros et al., 2023; Guastavino et al., 2022; Kashinath et al., 2021).

This article aims to evaluate whether deep learning models can provide skilful forecasts of extreme weather, by providing a pragmatic comparison between physics-based and data-driven models in a semi-operational setting. Specifically, it compares the performance of ECMWF's IFS HRES and leading global deep learning models in the task of forecasting near-surface temperature and windspeed extremes 1–10 days ahead, when provided with the same set of inputs, namely the output of IFS HRES at time 0. To do so, it makes use of the freely available forecast data provided by ECMWF and the WeatherBench 2 dataset (Rasp et al., 2024). The methods for the comparisons between models are largely based on the guidelines for evaluation of tail performance provided by Watson (2022), namely: i) comparison in terms of a standard metric (RMSE) computed on data beyond extreme quantiles only; ii) visual assessment of performance on extremes for specific regions/grid points; and iii) quantile-quantile plots of extreme quantiles to identify possible inconsistencies in tail estimation. All comparisons are performed at multiple time-scales (1-10 days) and for the whole globe, with separate metrics for each region following the ECMWF operational scorecards (ECMWF, 2024).

In the next two sections, we provide an introduction to the models included in the evaluation and the methods employed for the comparison. Then, we outline the results of the comparison for all the variables and regions of interest. Lastly, we reflect on the results of these comparisons, and on how they may affect the operational implementation of data-driven models. Additional results for models using ERA 5 reanalysis data (Hersbach et al., 2020) as input are included in Appendix D.

## 2 Models and Methodology

The rationale behind the choice of models and the methodology employed is to make the comparison between data-driven models and physics-based models as fair as possible. For this reason, we include in the main text only those data-driven models included in the WeatherBench 2 that are able to take the same set of initial conditions as IFS HRES, ECMWF's high-resolution deterministic forecasting system. All the models in the main take therefore as an input IFS HRES at time 0, and are able to produce 6 hourly forecasts of 2m temperature and 10m wind, the variables on which the models are evaluated. Those outputs are in turn all compared to the same ground truth, ERA 5 (Hersbach et al., 2020), at 1.5 degrees horizontal resolution, as in the WeatherBench 2 (Rasp et al., 2024). Indeed, models taking as input reanalysis data present a conceptual difference to operational models, as they are based on input data that is available with a considerable time delay and thus cannot be used in an operational setting.

Two-data driven models fit the criteria established above: operational Pangu-Weather (Bi et al., 2023) and operational Graph-Cast (Lam et al., 2023). We believe these models may represent reasonably well the performance of deterministic data-driven models as a whole, since they display similar performance to other data-driven models in a range of atmospheric and surface variables at multiple lead times (Rasp et al., 2024). Furthermore, these models employ the two leading architectures for deterministic data-driven weather forecasting, namely vision transformers (Dosovitskiy et al., 2020) and graph neural networks (Scarselli et al., 2009), respectively. Yet, recognising that some subtle differences may be lost by not including a more diverse range of data-driven models in our comparison, we present in Appendix D a comparison between IFS HRES and reanalysis-based deep learning models, namely reanalysis-based Pangu-Weather, reanalysis-based GraphCast, and FuXi (Chen et al., 2023b). These are currently regarded as the best deterministic data-driven models in terms of RMSE for medium to long range forecasting (Rasp et al., 2024).

In this section, we first provide a brief description of each of the models included in the comparison in the main, and then outline the criteria on which the comparison is based. For a complete description of the models including a full list of inputs and outputs, we refer the reader to Rasp et al. (2024) and Olivetti and Messori (2024), as well as to the original papers introducing the models described in Subsections 2.1-2.3.

## 2.1 IFS HRES

IFS HRES is ECMWF's flagship deterministic high-resolution model, widely regarded as one of the best physics-based numerical weather forecast models in the world (Rasp et al., 2020, 2024). All the parameters included in the model as well as its regular updates and improvements are thoroughly documented on ECMWF's website (Blanchonnet, 2022). Currently, IFS HRES takes a much larger set of inputs than any of the data-driven models, and also produces hourly forecasts for a very large set of outputs, at a $0.1°$ horizontal resolution on 137 pressure levels. The set of inputs forming IFS HRES's initial conditions (IFS HRES at time 0) are a mix of in-situ observations for the three hours surrounding the forecast and model outputs from the previous IFS HRES run. IFS HRES is included here as baseline to which to compare the performance of data-driven models. All IFS HRES forecasts have been generated with the operational version of the model used at the time of the forecast (Rasp et al., 2024), namely model configuration Cy46r1 for forecasts initiated before 2020-06-30, and Cy47r1 for forecasts initiated after that date.

## 2.2 Pangu-Weather

Pangu-Weather (Bi et al., 2023) is a data-driven, deep learning model using a vision transformer architecture (Dosovitskiy et al., 2020). First developed in 2022 (Bi et al., 2022) and published in 2023 (Bi et al., 2023), it is the "oldest" data-driven model among those included in the comparison. It is trained on ERA5 reanalysis data for 1979 to 2017 and uses 2018-2019 as validation. It takes as input five upper-air variables on thirteen atmospheric levels and four surface variables, and it produces forecasts of those same variables for the next atmospheric state 6-hours ahead, in a sequential manner. The output of the model can then be fed again as input, to obtain forecasts at longer lead times. In this way, it is possible to obtain forecasts up to 10 days ahead, at $0.25°$ resolution. In its operational version, analysed in the main text here, Pangu-Weather takes as input IFS HRES at time 0, while in the version included in Appendix D it takes ERA 5 as initial state. The operational and reanalysis-based versions of Pangu-Weather are otherwise identical.

## 2.3 GraphCast

GraphCast (Lam et al., 2023) is a deep learning model using a graph-based architecture (Scarselli et al., 2009). First developed in late 2022 (Lam et al., 2022) and published in Lam et al. (2023), it builds on earlier work by Keisler (2022). It is trained on ERA5 reanalysis data for 1979 to 2019 and, in the operational version, additionally fine-tuned on a smaller sample of IFS HRES data. It takes as input six atmospheric variables at 37 atmospheric levels, and numerous surface variables and masks. GraphCast aims to forecast the next state of the atmosphere as a function of its two previous states, in a sequential manner. As for Pangu-Weather, it produces 6-hourly forecasts up to 10 days ahead, at $0.25°$ resolution. The main difference between the operational version analysed in the main text and the version included in Appendix D is that the operational version does not require precipitation as input, thus allowing the use of IFS HRES at time 0 as input.

## 2.4 Criteria for model comparison

The comparison between models is based on their performance in forecasting 2m temperature cold and hot extremes and 10m windspeed extremes globally. Following the WeatherBench 2 (Rasp et al., 2024), the models are tasked with forecasts with a timestep of 6h or less, and all comparisons are based on a spatial resolution of 1.5 degrees. Forecasts are initiated every 12 hours (00:00 and 12:00) for the period 01-01-2020 to 16-12-2020, thus providing 702 comparable forecasts for each lead time and grid point. Comparisons are performed globally, and for regions included in the ECMWF operational scorecards (ECMWF, 2024), as defined in Table 1:

**Table 1.** Regions for forecast performance evaluation, in accordance with ECMWF's operational scorecards (ECMWF, 2024)

| Region | Definition |
|---|---|
| Northern Hemisphere (Extra-tropics) | lat $\geq 20°$ |
| Southern Hemisphere (Extra-tropics) | lat $\leq$ -20° |
| Tropics | -20° $\leq$ lat $\leq$ 20° |
| Extra-tropics | \|lat\| $\geq$ 20° |
| Arctic | lat $\geq 60°$ |
| Antarctic | lat $\leq$ -60° |
| Europe | 35° $\leq$ lat $\leq$ 75°, -12.5° $\leq$ lon $\leq$ 42.5° |
| North America | 25° $\leq$ lat $\leq$ 60°, -120° $\leq$ lon $\leq$ -75° |
| North Atlantic | 25° $\leq$ lat $\leq$ 60°, -70° $\leq$ lon $\leq$ -20° |
| North Pacific | 25° $\leq$ lat $\leq$ 60°, 145° $\leq$ lon $\leq$ -130° |
| East Asia | 25° $\leq$ lat $\leq$ 60°, 102.5° $\leq$ lon $\leq$ 150° |
| AusNZ | -45° $\leq$ lat $\leq$ -12.5°, 120° $\leq$ lon $\leq$ 175° |

For the sake of conciseness, we focus our comparison here on forecasts for 1, 3, 5, 7 and 10 days ahead. We evaluate the performance of the models based on three different criteria, largely based on the recommendations for evaluation of extreme event forecasts provided by Watson (2022). The criteria are as follows:

1. Accuracy in determining the magnitude of the most extreme data-points globally or within a given region. To define the extremes, we pool together all data-points for 2020 for the region of choice, and set a threshold based on a quantile of choice out of all the data-points. We then consider as extreme all data-points exceeding that threshold. Thus, we allow for any number of global and regional extremes to come from a specific grid point or time. The number of data-points used for evaluation in each region becomes then 702 (data-points at each grid point) multiplied by the number of grid points within the specific region and by the percentage of data-points exceeding the chosen quantile-based threshold. For example, if the top 5% of events are considered, the number of data-points for evaluation would be 702 times the number of grid points in the region times 0.05.

Accuracy is measured in terms of RMSE (lower values are better), as defined below:

- For hot and windspeed extremes:

$$RMSE_t = \sqrt{\frac{1}{TIJ}\sum_t^T\sum_i^I\sum_j^J w(i)\mathbb{1}_{o_t>Q(o)}(\hat{y}_{t,i,j}-o_{t,i,j})^2} \tag{1}$$

- For cold extremes:

$$RMSE_t = \sqrt{\frac{1}{TIJ}\sum_t^T\sum_i^I\sum_j^J w(i)\mathbb{1}_{o_t<Q(o)}(\hat{y}_{t,i,j}-o_{t,i,j})^2}, \tag{2}$$

where,

$1,2,3,...,T$ is the available number of time-points at the given forecast lead time. $T$ is 702 in our case;

$1,2,3,...,I$ is the number of points of latitude included in the region of interest,

$1,2,3,...,J$ is the number of points of longitude included in the region of interest,

$\hat{y}$ is the forecasted value of the variable of interest,

$o$ is the observed value of the variable of interest, in our case from ERA5,

$\mathbb{1}_{o_t>Q(o)}$ is an indicator function taking a value of 1 for data-points above the chosen quantile of the variable of interest in the given region, and 0 otherwise. For cold extremes, $\mathbb{1}_{o_t<Q(o)}$ so that the indicator function takes a value of 1 for data-points

below the chosen quantile, and 0 otherwise. Differences in performance between models are assessed for significance at the 5% level by using a paired t-test with cluster-robust standard errors (Liang and Zeger, 1986; Arellano, 1987; Cameron and Miller, 2015), which accounts for the spatial and temporal clustering of extreme events. The test is conducted two-sided when comparing data-driven models to IFS HRES, and one-sided when specifically assessing whether the best individual model significantly outperforms the second-best within a specific region.

2. Accuracy in determining the magnitude of grid-point extremes. Extremes are defined as in criterion 1, but at a grid point level, by defining a different threshold and set of extremes for each grid point. The RMSE is computed according to Equations 1 and 2, with a redefined indicator function. For hot extremes, the indicator function is given by $\mathbb{1}_{o_{t,i,j}\geq Q(o_{i,j})}$, taking the value of 1 for data-points above or equal to the quantile of interest at the given point of latitude and longitude, and 0 otherwise. For

cold extremes, the indicator function becomes $\mathbb{1}_{o_{t,i,j}\leq Q(o_{i,j})}$. Thus, the number of data-points available at each grid-point is

702 multiplied by the percentage of data-points exceeding the chosen quantile.

Grid-point level differences in performance between the best data-driven model and IFS HRES are assessed for significance using the same approach as for point (1) above. The obtained p-values are corrected for multiple testing by applying global false discovery rates (Benjamini and Hochberg, 1995; Wilks, 2016) using a global significance level of 0.1. This corresponds to an approximate significance level of 0.05 in the presence of strongly spatially correlated events (Wilks, 2016), such as near-surface temperature extremes.

3. Calibration of extreme quantiles, where a quantile behaviour closer to the ground-truth (ERA 5) is considered superior to a quantile behaviour further away from it. We evaluate extreme quantile behaviour by considering quantiles between 90 and 99.9 for hot and wind extremes, and quantiles between 10 and 0.1 for cold extremes. We then produce quantile-quantile plots, where the extreme quantiles in the forecasts are plotted against the corresponding quantiles of ERA 5.

The three criteria jointly provide an overall picture of the performance of the models at forecasting near-surface temperature and wind extremes at global and regional (criterion 1), and local level (criterion 2), as well as of the tail behaviour of the models when faced with values at the edges or beyond the limits of the training distribution (criterion 3).

# 3 Results

In this section, we report the results of the model comparison performed according to the criteria outlined in Subsection 2.4. The aim here is both to provide a comparison between data-driven and physics-based models as a whole, as well as to identify relevant differences between the data-driven models themselves.

We start by providing an overview of the performance of different models globally and in individual regions when considering all data-points, both extremes and not extreme. For all models, performance differences between regions are small, especially for 10m windspeed (Figure 1). Both data-driven models perform significantly better than ECMWF IFS HRES globally and in most regions. Most impressively, GraphCast significantly outperforms IFS HRES for all regions and lead times.

The difference between GraphCast and Pangu-Weather is smaller overall, with the largest differences observed in 2-meter temperature forecasts at longer lead times. Notably, GraphCast consistently outperforms Pangu-Weather across all regions in these longer-range forecasts, with differences in the range of 5 to 20% for 10 days forecasts of 2m temperature. The strong performance of GraphCast may partly depend on its training scheme, which assigns additional weight to surface and lower tropospheric variables at the expense of higher atmospheric levels (Lam et al., 2023).

The data-driven models' performance, especially for Pangu-Weather, appears to deteriorate at a faster rate than IFS HRES at longer lead times. This might be a sign that data-driven models suffer from "blurring"(Bonavita, 2024; Price et al., 2024), namely the tendency to revert to the climatology and produce progressively less skilful forecasts with increasing lead time. While this problem applies to both physical and data-driven models, it has recently been shown to be prominent among data-driven models (Bonavita, 2024).

## RMSE scorecard based on all test data-points

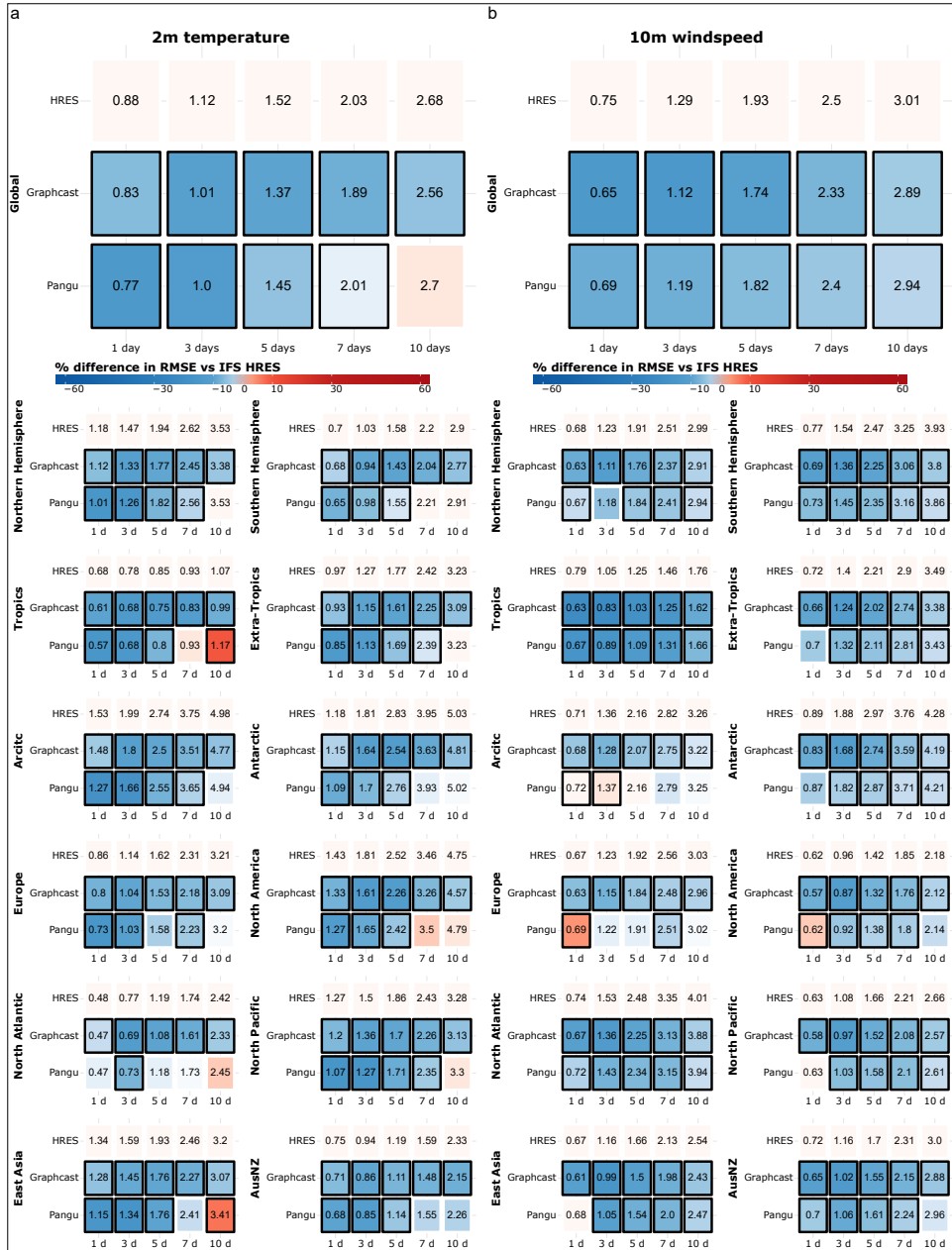

**Figure 1.** RMSE scorecard for 2m temperature (a) and 10m windspeed (b) at a global and regional scale, computed on all test data-points. Blue shades indicate better performance than IFS HRES, red shades worse performance. Black borders indicate significantly different performance from IFS HRES, at the 5% level.

Figure 2 provides RMSE comparisons for the 5% most extreme data-points globally and in each region, in accordance with criterion 1 (Subsection 2.4). Globally, GraphCast significantly outperforms IFS HRES for all three categories at most lead times, with the largest differences in terms of hot and windy extremes. Pangu-Weather performs more similarly to IFS HRES, with statistically significant improvements in performance only for hot extremes at shorter lead-times (1-3 days ahead), and worse performance than IFS HRES for hot and windy extremes at longer lead times.

Similar to what observed for all data, the performance on extremes of both data-driven models degrades compared to IFS HRES for longer forecast lead times. This is particularly notable for 10-days ahead predictions. This is perhaps not surprising, given that the 10-day predictions are close to the limits of skilful forecast for extremes. However, this may also be interpreted as an additional sign of blurring. The fact that the data-driven models build on the iterative feeding of the most recent atmospheric states into the models to generate a one-step-ahead forecast may also play a role in this respect. Indeed, this approach may contribute to the accumulation of small errors over time that become more relevant for extreme weather forecasts at longer lead times (Bonavita, 2024).

Regional comparisons between models largely confirm the above patterns, while also revealing some additional detail. Overall, data-driven models demonstrate better performance relative to IFS HRES in the Northern Hemisphere than in the Southern Hemisphere, particularly for cold extremes. Notably, IFS HRES significantly outperforms both data-driven models in AusNZ and Antarctica for cold extremes, and in East Asia, North America, and Antarctica for hot extremes. The comparatively poor performance of the data-driven models in Antarctica may depend on the lower quality of reanalysis data for this region, on which the data-driven models are trained.

Conversely, GraphCast outperforms IFS HRES in the tropics and the North Pacific for all variables at all lead times, and in the Arctic for temperature extremes. We speculate that some of these regional differences may depend on the lack of input variables relevant to near-surface extremes (e.g soil moisture and snow and ice cover) in the training of data-driven models, which might play a more prominent role in certain regions than others (e.g. soil moisture for hot extremes in continental North America and East Asia (Coronato et al., 2020; Liu et al., 2014)).

Additionally, we observe that in most regions data-driven models perform better for temperature than wind extremes relative to IFS HRES. A possible reason for this might be the lack of specific training on 10m windspeed for GraphCast and Pangu-Weather, which are, instead, trained on u- and v-wind components separately. This approach may be suboptimal for windspeed extremes, as the non-linear relationship between errors in the individual wind components and the resultant total windspeed can lead to large errors in windspeed forecasts. Even a small underestimation in one wind component can result in a substantial underestimation of total windspeed under strong wind conditions.

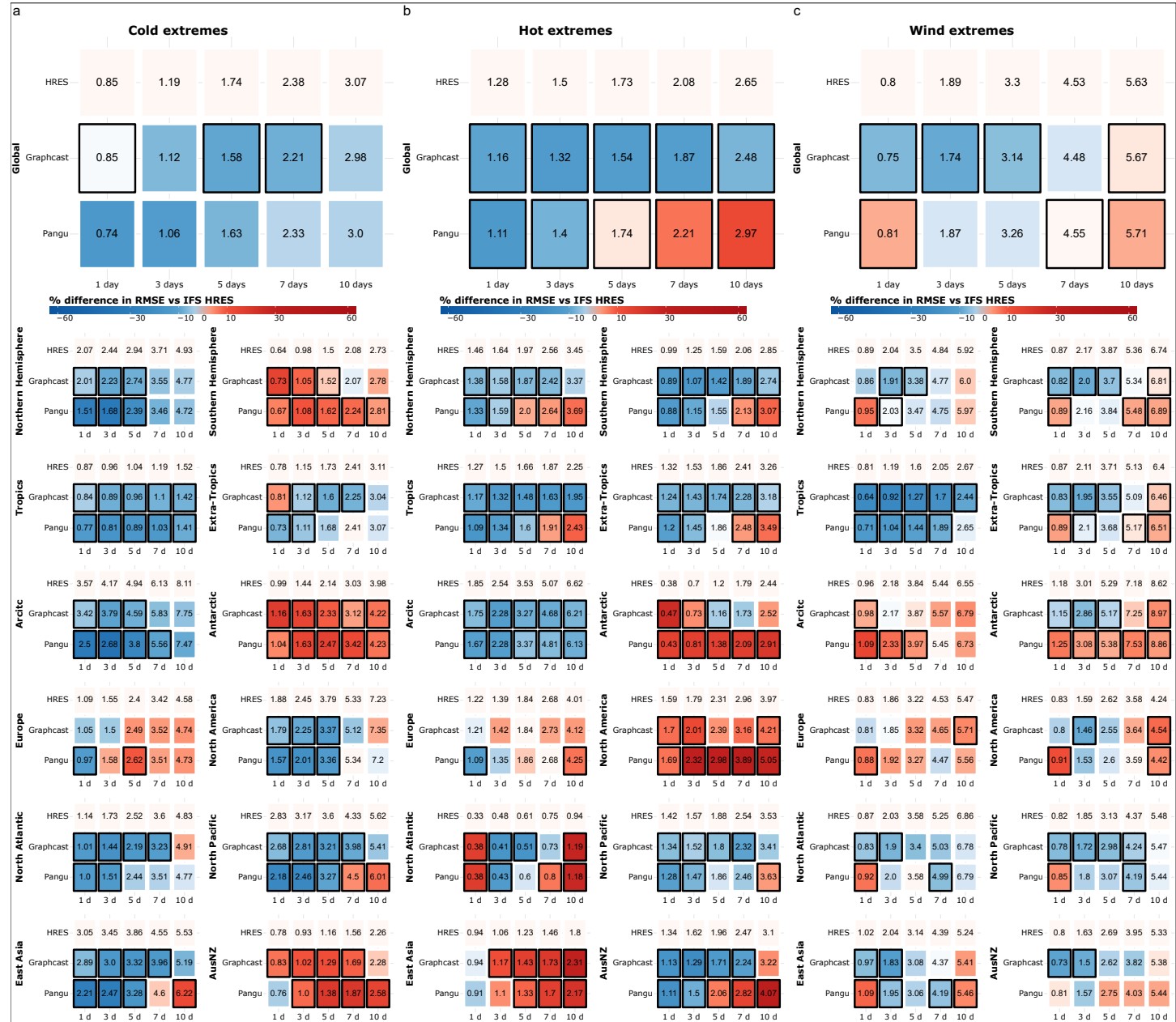

**Figure 2.** RMSE scorecard for cold (a), hot (b) and wind extremes (c) at a global and regional scale, computed on the (a) 5% lowest 2m temperature, (b) 5% highest 2m temperature and (c) 5% highest 10m windspeed data-points, respectively. Black borders indicate statistically significant differences in performance from IFS HRES, at the 5% level.

Figure 3 repeats the analysis presented in Figure 2, but for the 1% most extreme data-points in each region. The conclusions drawn from Figure 3 largely hold for wind and hot extremes, but not necessarily for cold extremes. Specifically, there is a noticeable decline in the performance of data-driven models, particularly GraphCast, for cold extremes both globally and in the Extra-Tropics. However, it is important to consider that our approach to selecting extremes may result in a higher proportion of global cold extremes originating from Antarctica in Figure 3 than in Figure 2. This could explain the worse performance of data-driven models, given their relatively weak performance in this region.

Additionally, we observe larger regional differences in Figure 3 than in previous figures. This may be the result of the smaller sample size, as well as of the larger variability associated with a smaller number of extreme events. The difference in performance between the Northern and Southern Hemispheres becomes more evident for cold events, while hemispheric differences for hot and windy events are often not statistically significant and largely dependent on lead time. Notably, we observe significant performance differences for cold extremes in East Asia at shorter lead times, where Pangu-Weather outperforms other models by up to 40%. For hot extremes, IFS HRES significantly outperforms data-driven models in North America, East Asia, and AusNZ at longer lead times, while Pangu-Weather is clearly outperformed by other models. The strong performance of Pangu-Weather for cold extremes, combined with its weaker performance for hot extremes, suggests a possible cold bias in some regions.

In terms of regional wind extremes, few results are statistically significant, and those that do mostly confirm performance patterns already discussed: notably, data-driven models outperform the physical model in the tropics, and tend in general to show better performance at short rather than long lead times. The strong performance of data-driven models in the tropics for both the top 5% and 1% of events may be related to the use of latitude-based weights, which drive best performance towards the Equator. In terms of mid-latitudes performance, the three models perform overall similarly, with differences between models being mostly lead-time dependent and rarely statistically significant.

# RMSE scorecard for 1% most extreme data-points

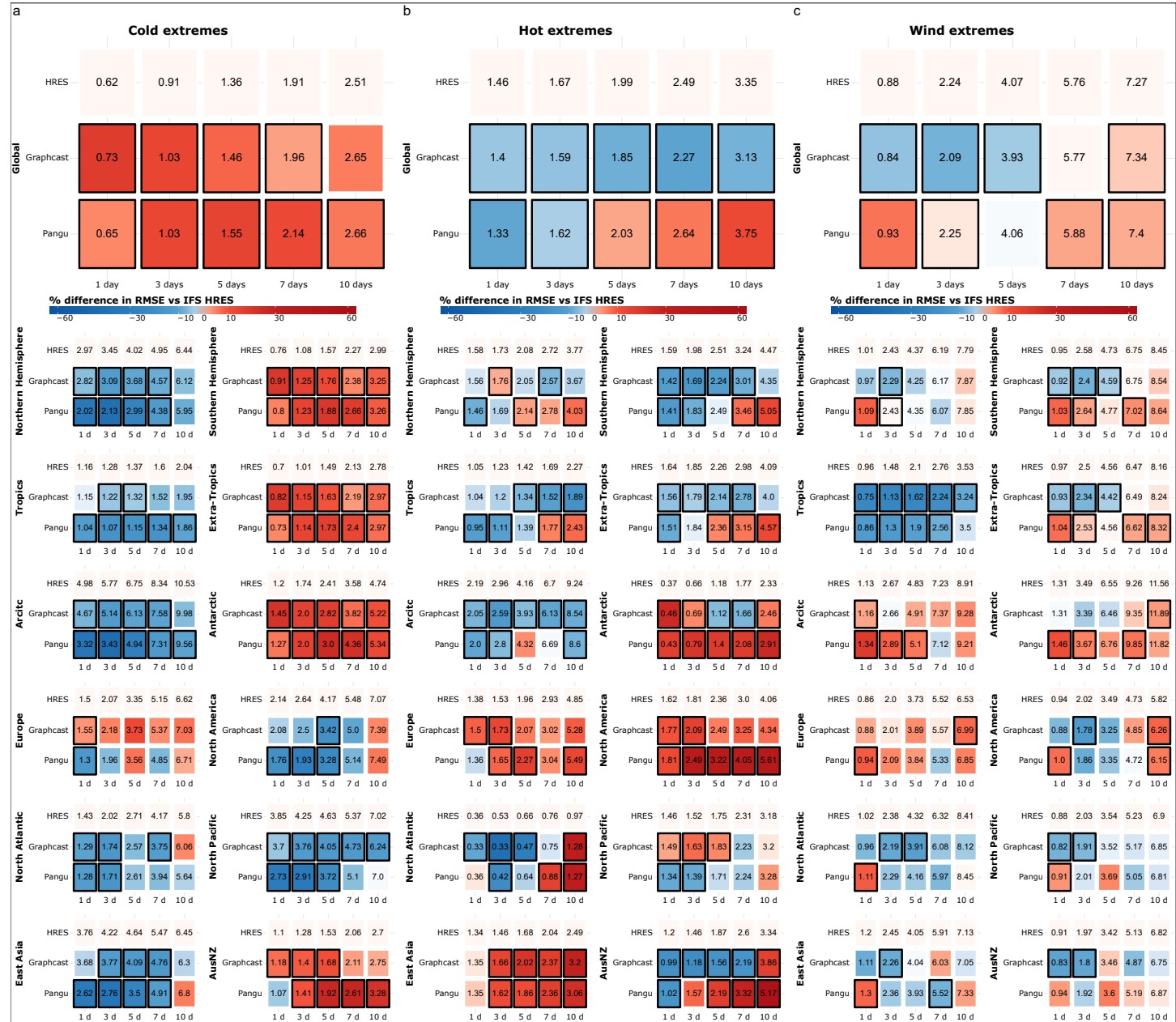

**Figure 3.** RMSE scorecard for cold (a), hot (b) and wind extremes (c) at a global and regional scale, computed on the (a) 1% lowest 2m temperature, (b) 1% highest 2m temperature and (c) 1% highest 10m windspeed data-points, respectively. Black borders indicate statistically significant differences in performance from IFS HRES, at the 5% level.

A summary scorecard of Figures 1–3 is provided in Figure 4, showing which of the three models is best at forecasting cold, hot and windspeed extremes as well as 2m temperature and 10m windspeed overall. The summary scorecard confirms the patterns observed so far, suggesting that data-driven models are generally superior to IFS HRES at forecasting 10m wind and 2m temperature when considering all data-points. However, the summary scorecard also shows that the performance of data-driven models degrades relative to IFS HRES when considering extreme quantiles, with IFS HRES being overall superior at forecasting cold extremes in AusNZ and Antarctica, and mostly outperforming data-driven models in forecasting hot extremes in Europe, North America and East Asia. Nevertheless, IFS HRES and data-driven models display comparable performance in forecasting other types of extremes in those regions. Additionally, the summary scorecard highlights the progressive deterioration in performance of data-driven models compared to IFS HRES for extremes at longer lead times, likely connected to the above-mentioned blurring.

**Figure 4. a-f**: Best model in terms of tail-RMSE computed on the (a) 5% lowest 2m temperature, (b) 5% highest 2m temperature, (c) 5% highest 10m windspeed, (d) 1% lowest 2m temperature, (e) 1% highest 2m temperature, (f) 1% highest 10m windspeed data-points. **g-h** Best model in terms of overall RMSE for (g) 2m temperature and (h) 10m windspeed. Black borders indicate statistically significant better performance than the other models, at the 5% level.

Figure 5 and Figure 6 apply criterion 2 (Subsection 2.4) to the comparison between models, to evaluate grid-point level differences in RMSE between IFS HRES and the best data-driven model for that grid point. The data-driven models are better than IFS HRES in terms of overall RMSE in most locations, with the only exception of 1 day 2m temperature forecasts, where the performance of the models is highly latitude dependent. This latitude-dependent pattern can be observed, to a lesser extent, even in all other subfigures, where data-driven models consistently perform at their best close to the tropics while displaying a performance more similar to IFS HRES in the extra-tropics. This supports the above-mentioned thesis that the latitude-based weights used by the data-driven models may drive best performance towards low-latitude areas (see also Figure 2 and 3).

Figure 6 provides complementary information by highlighting the magnitude of the differences between models, independent of their statistical significance. As in previous cases, data-driven models become progressively worse compared to IFS HRES at longer lead times, further supporting the above-mentioned blurring thesis. Moreover, we notice, especially for temperature extremes, a tendency for data-driven models to perform better on the west side of the Pacific and Atlantic Ocean, and worse on the east side. While this pattern is not as evident as the latitude- and lead-time dependent performance, it is likely tied to the lack of information on ocean processes and sea-surface temperatures as an input to data-driven models. This omission may, for instance, lead to underestimating the effects of underwater currents and upwelling in certain regions, where these processes play an important role in defining local climates (e.g. Abrahams et al., 2021; Jacox et al., 2015; Lemos and Pires, 2004). The lack of information on sea-surface temperatures might also be connected to the subpar performance of data-driven models for 10m windspeed in some specific areas within the intertropical convergence zone, such as in the DRC and North-western South America (Chiang et al., 2002).

**RMSE pixel by pixel - which model is best?**

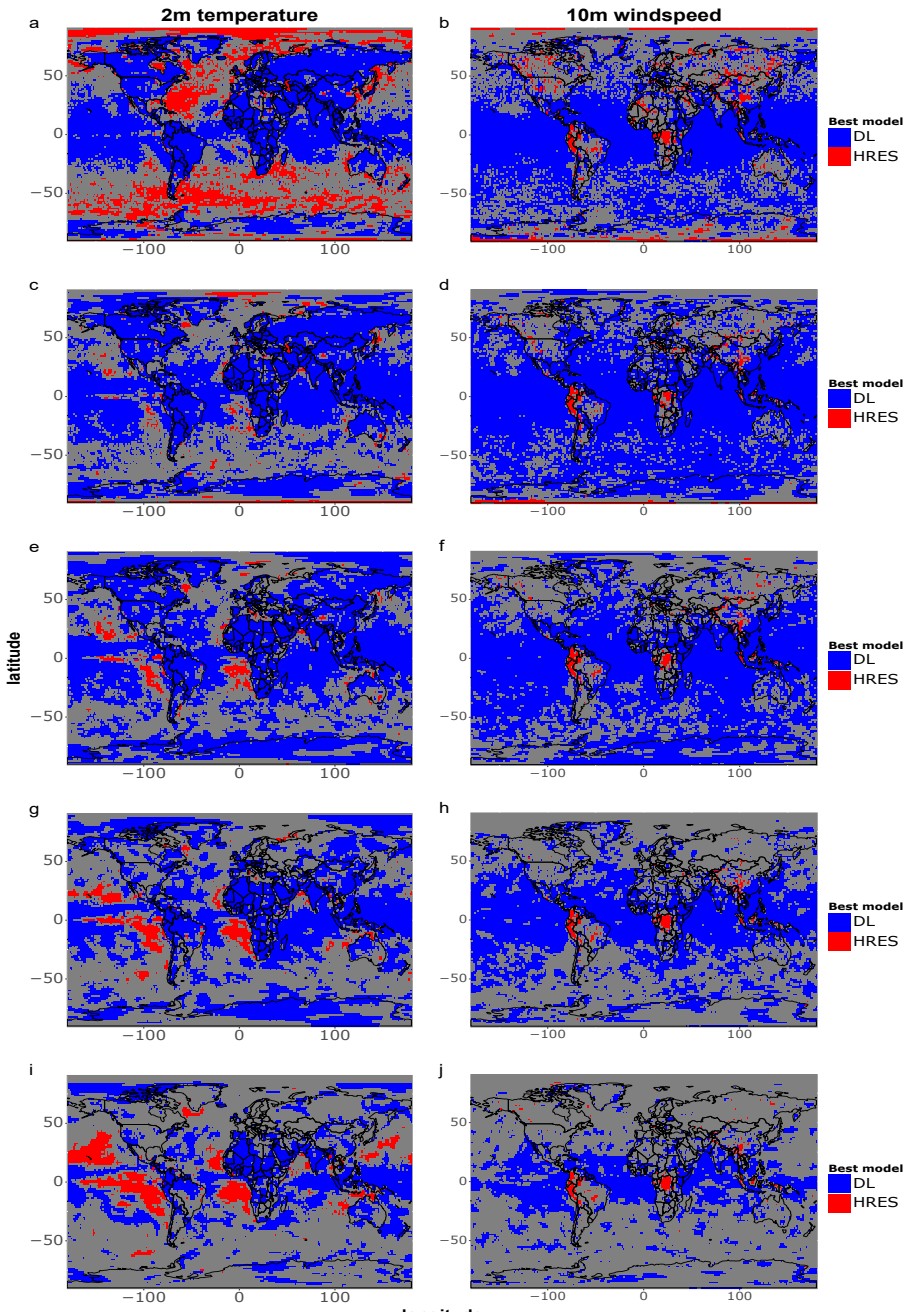

**Figure 5.** Single-gridpoint RMSE comparison for all data-points of 2m temperature and 10m windspeed. Blue shades indicate that the best data-driven, deep learning model (DL) at that grid point is significantly better than IFS HRES at the 5% level, while red shades indicate that IFS HRES is better. Gray shades indicate no statistically significant differences. a-b) 1 day forecasts; c-d) 3 days forecasts; e-f) 5 days forecasts; g-h) 7 days forecasts; i-j) 10 days forecasts.

**RMSE pixel by pixel - magnitude of differences**

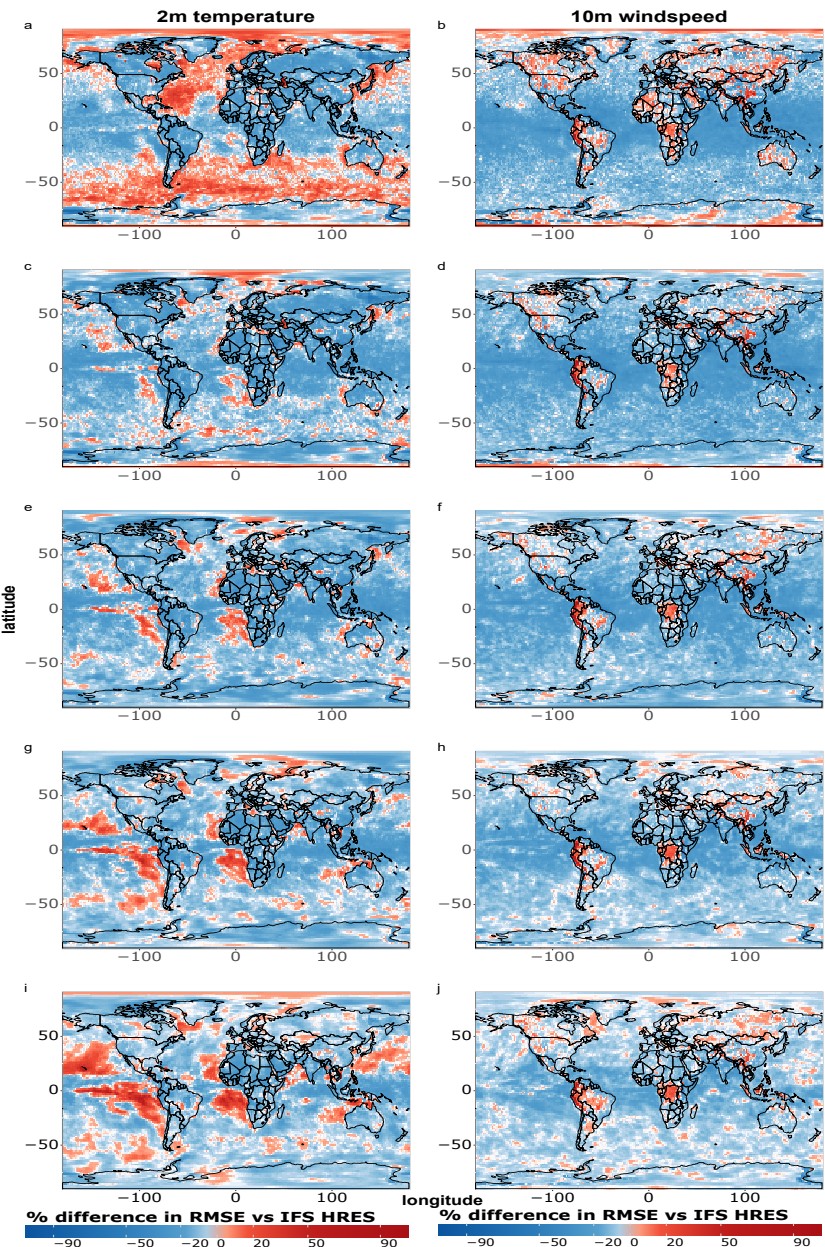

**Figure 6.** Magnitude of single-grid point RMSE differences between IFS HRES and the best data-driven model at that each grid point for all data-points of 2m temperature (a) and 10m windspeed (b). Blue shades indicate better performance for the data-driven model, while red shades indicate better performance by IFS HRES. a-b) 1 day forecasts; c-d) 3 days forecasts; e-f) 5 days forecasts; g-h) 7 days forecasts; i-j) 10 days forecasts.

Figure 7 and Figure 8 correspond to figures 5 and 6 but for temperature and wind events exceeding the 5% most extreme quantile of the respective distributions at the given grid point during the test period. Fewer differences between models are statistically significant when looking specifically at extremes, likely due to the smaller sample size (n=36) as well as the fact that IFS HRES and the data-driven models perform more similarly overall. We observe, in particular, only few significant differences between IFS HRES and the data-driven models for windspeed extremes, where the high variance in the magnitude of the windspeed extremes may affect the size of the test statistic and prevent achieving statistical significance even in the presence of large absolute differences in performance.

Despite this, it is still possible to identify some clear patterns. Once more, data-driven models perform best in the tropics as a whole, and worse closer to the poles. This is particularly true for hot extremes, where IFS HRES clearly outperforms data-driven models near the arctic and in vast ocean areas in the Southern-extratropics. This is largely in line with what found in Figures 1–4, and likely ascribable to the same reasons. Additionally, also in line with what previously found in the above-mentioned plots, we find evidence of blurring, especially for cold extremes.

Examining the magnitude of differences between the models (Figure 8), we observe significant discrepancies primarily near the poles and over the oceans in terms of temperature extremes. Specifically, for hot extremes, data-driven models tend to perform worse on the eastern sides of ocean basins, consistent with what found in Figure 6. Regarding windspeed extremes, the poorer performance of data-driven models overall may again be attributed to the lack of separate training for u- and v-wind components, which can lead to amplified errors for extremes. Additionally, we observe that IFS HRES consistently outperforms data-driven models in many densely populated regions, including parts of the US, China, and Northern India. Although these differences are mostly not statistically significant, they nonetheless highlight the need for caution when considering the operationalisation of data-driven models for forecasting windspeed extremes.

## RMSE pixel by pixel - which model is best?

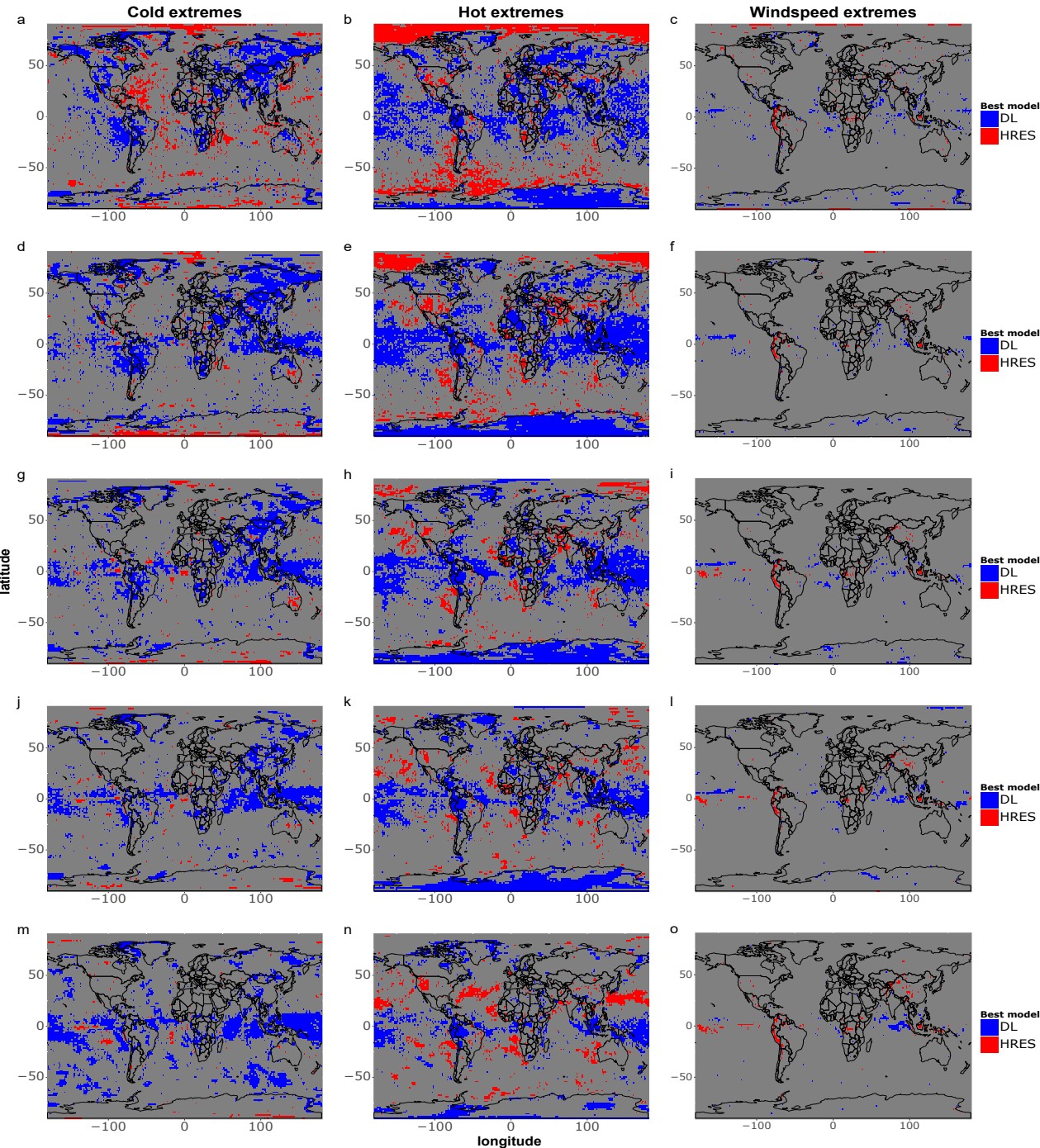

**Figure 7.** As in Figure 5 but for cold, hot and windspeed extremes. The extremes are defined as in Figure 2, but for individual grid points. a-c) 1 day forecasts; d-f) 3 days forecasts; g-i) 5 days forecasts; j-l) 7 days forecasts; m-o) 10 days forecasts. The number of data-points per grid-point is 36.

**RMSE pixel by pixel - magnitude of differences**

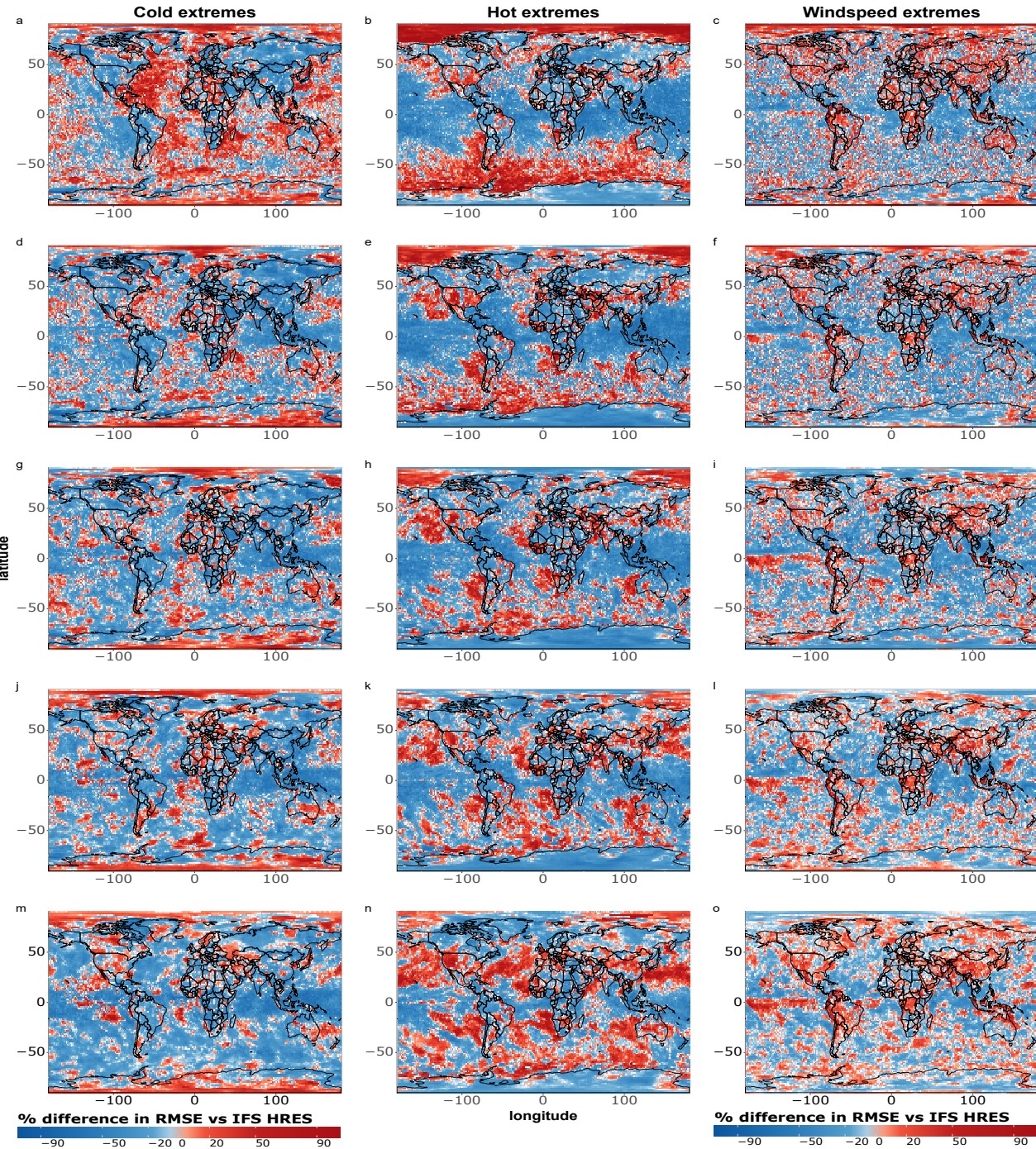

**Figure 8.** As in Figure 6 but for cold, hot and windy extremes. The extremes are defined as in Figure 2, but for individual grid points. a-c) 1 day forecasts; d-f) 3 days forecasts; g-i) 5 days forecasts; j-l) 7 days forecasts; m-o) 10 days forecasts. The number of data-points per grid-point is 36.

Lastly, we compare the models on the basis of criterion 3 (Subsection 2.4), namely on the ability of different models to reproduce the tail behaviour of ERA 5. As in the previous cases, we start by looking at global extremes at multiple lead times (Figure 9), in order to assess the tail behaviour of the forecasts exceeding the 10% most extreme quantile of their respective distributions. Figure 9 suggests that all models appear to be well calibrated in the forecast of global cold extremes, while data-driven models tend to underestimate the magnitude of hot and windspeed extremes, especially at longer lead times. The increasing underestimation of extremes of data-driven models at longer lead times is in line with previous findings in this paper related to blurring.

As in previous cases, regional patterns reveal further complexities in the behaviour of the three models. Figure 10 suggests that all models tend to underestimate cold extremes in the Arctic and North Pacific. Additionally, data-driven models tend to underestimate cold extremes in the Antarctic, and IFS HRES and GraphCast also in Europe. The largest underestimation occurs in the Arctic, with the coldest data-points being underestimated by 2-3 K by all models, on average. This is in line with the previous findings suggesting that data-driven models struggle more with extreme forecasts further away from the tropics. Moreover, we find that the underestimation of cold extremes is in many cases more severe for GraphCast than for Pangu-Weather, reinforcing the impression that Pangu-Weather might have a cold bias compared to GraphCast.

This thesis is also supported by Figure 11, which, conversely, shows a more severe underestimation of hot extremes for Pangu-Weather and better tail reliability for GraphCast. However, even in this case IFS HRES displays the best tail reliability overall, whereas both data-driven models tend to underestimate extremes in several regions, including North America, East Asia, Europe, the Tropics and the North Pacific. AusNZ appears to be the only region where some of the models (IFS HRES and GraphCast) overestimate the average magnitude of the extremes, a finding for which we do not find an immediate explanation. Once more, data-driven models seem to suffer from more severe lack of calibration for regions further away from the Equator, with the largest underestimations occurring in North America, where the data-driven models underestimate the warmest data-points by around 2 K, on average.

Similar to temperature extremes, IFS HRES displays almost perfect tail behaviour for wind extremes (Figure 12), whereas data-driven models tend to slightly underestimate windspeed extremes in all regions. The differences between models, and especially GraphCast and Pangu-Weather are, however, smaller overall. The largest difference in tail reliability between Graph-Cast and Pangu-Weather is in the Tropics, where, as found in Figures 2 and 3, GraphCast appears to outperform Pangu-Weather.

**QQ plots 10% most extremes values globally**

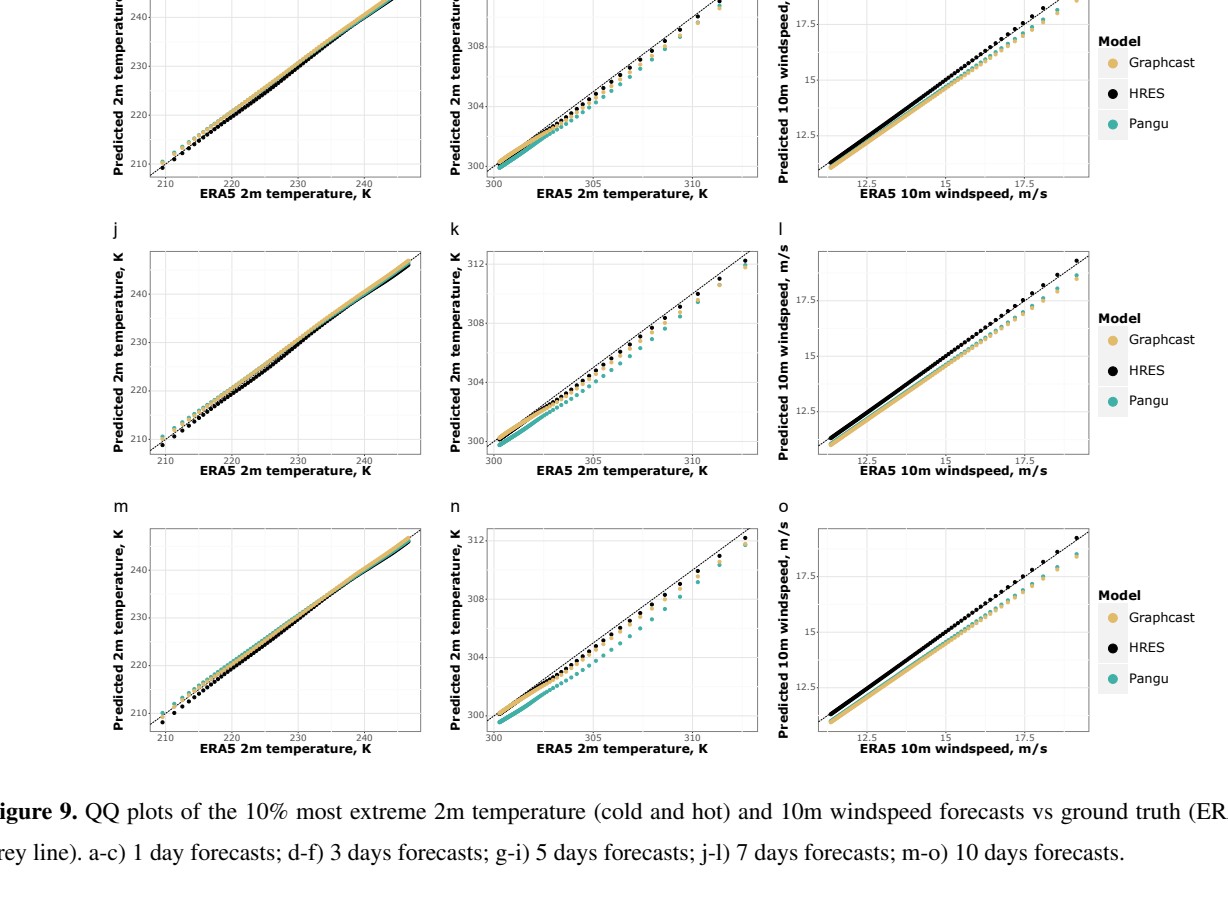

**Figure 9.** QQ plots of the 10% most extreme 2m temperature (cold and hot) and 10m windspeed forecasts vs ground truth (ERA 5, solid grey line). a-c) 1 day forecasts; d-f) 3 days forecasts; g-i) 5 days forecasts; j-l) 7 days forecasts; m-o) 10 days forecasts.

# Regional QQ plots cold extremes

**Figure 10.** Regional QQ plots of 5-day forecasts for the 10% coldest data-points in terms of ERA 5 2m temperature.

**Regional QQ plots hot extremes**

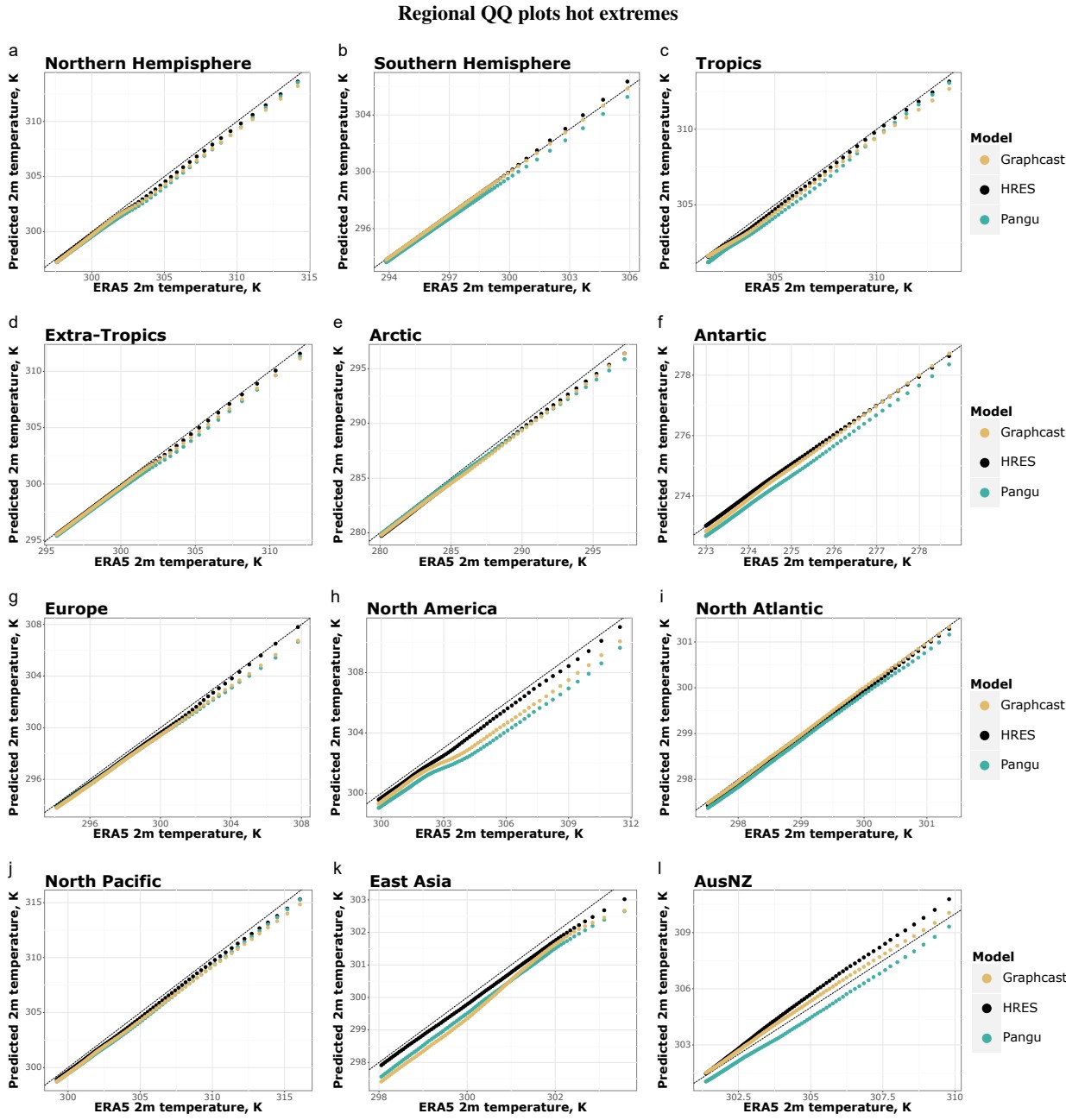

**Figure 11.** As in Figure 10 but for the 10% hottest data-points in terms of ERA 5 2m temperature.

## Regional QQ plots wind extremes

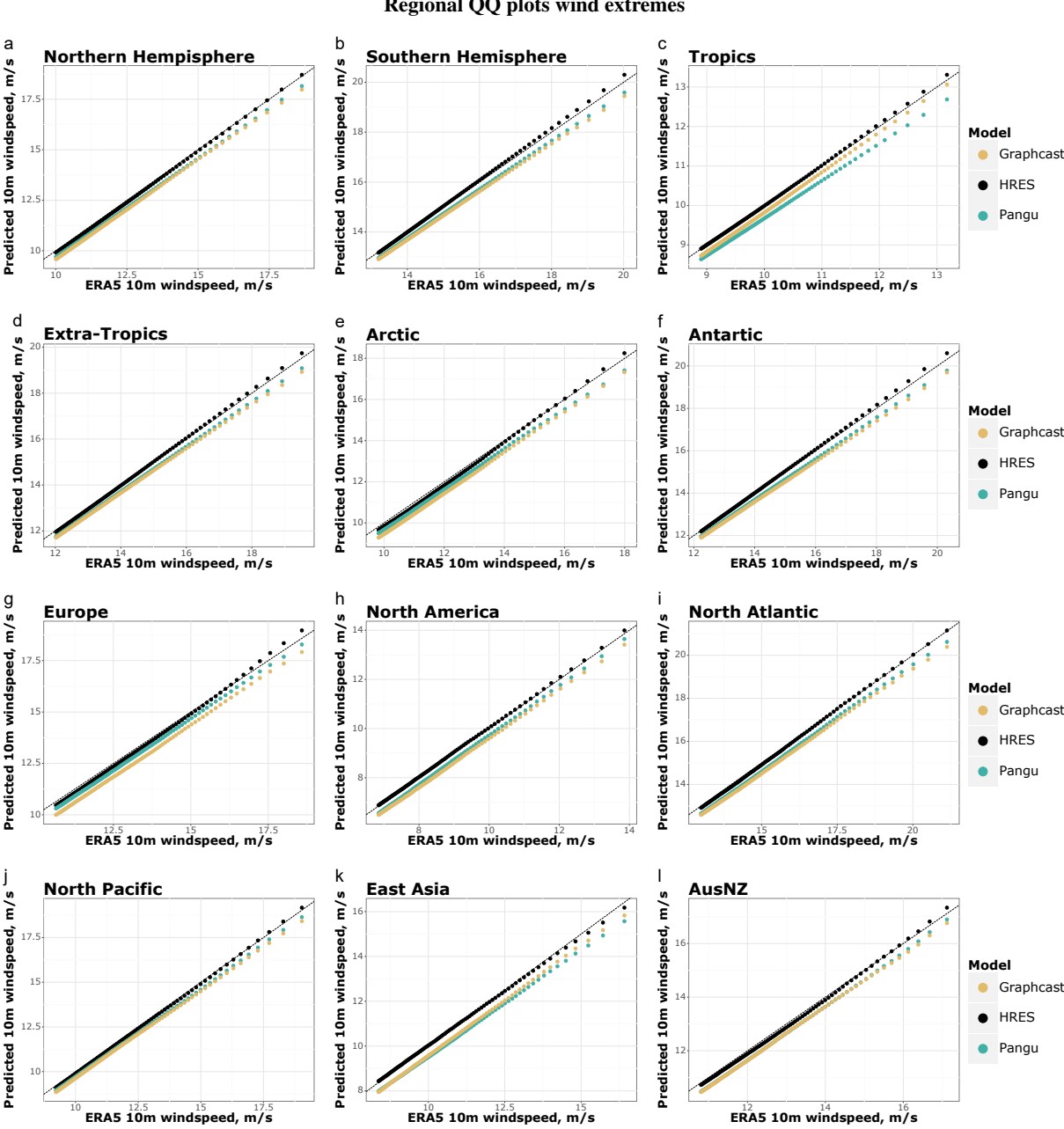

**Figure 12.** As in Figure 10 but for the 10% windiest data-points in terms of ERA 5 10m windspeed.

## 4 Discussion and Conclusions

This paper analyses the performance of ECMWF IFS HRES, GraphCast and Pangu-Weather in forecasting near-surface temperature and windspeed extremes up to 10 days ahead in a semi-operational setting. Following Watson (2022), the models have been evaluated with the help of three criteria (Subsection 2.4), assessing the forecast performance (criteria 1 and 2) and the calibration of the forecasts in the tails of the distribution (criterion 3). The results suggest that data-driven models are superior to IFS HRES in the task of forecasting 2m temperature and 10m windspeed on average in most regions (Figure 4), and especially in the Tropics (Figures 1, 5). Some notable exceptions include the eastern side of ocean basins for 2m temperature, and selected areas within the intertropical convergence zone for 10m windspeed. The weaker performance of data-driven models in these areas might depend on the lack of information related to ocean dynamics, and the omission of sea-surface temperature among their input variables.

In terms of extremes, the performance of data-driven models and IFS HRES is comparable overall, especially in terms of 10m windspeed (Figure 7). For temperature extremes, data-driven models mostly outperform IFS HRES in the tropics, while displaying a comparatively weaker performance at higher latitudes. Throughout our evaluation, we observe a pronounced meridional behaviour in the quality of data-driven forecasts, with a gradual deterioration of performance towards higher latitudes. We speculate this may partly depend on the use of latitude-based weights in the training of the data-driven models, which pushes them towards minimisation of large errors closer to the Equator at the expense of performance at higher latitudes.

Our results for 10m windspeed provide additional arguments to motivate caution in the operationalisation of data-driven models. IFS HRES outperforms the data-driven models in several densely populated land-areas, including Europe, the US and South-East Asia (Figure 8). This may partially depend on the stronger spatial heterogeneity of extremes over land regions, where the larger number of variables and physics-based framework of IFS HRES provide an advantage. The overall weaker performance of data-driven models for windspeed extremes compared to temperature extremes may also depend on the separate training of u-and v-wind components employed by Graphcast and Pangu-Weather.

A more general finding is that the data-driven models perform best in relative terms at shorter lead times, whereas IFS HRES performs best in relative terms at longer lead times (Figures 1– 3). We tie this behaviour to the phenomenon of blurring, which has been highlighted as a problem faced by deterministic data-driven models in recent studies (Bonavita, 2024; Price et al., 2024). As lead time and uncertainty increase, data-driven models tend to revert to the climatology to minimise large errors. Although this behaviour is common to all weather models, it is more pronounced in deterministic data-driven models compared to numerical models. However, probabilistic data-driven models, which are currently under development (e.g. Price et al., 2024; Lang et al., 2024; Oskarsson et al., 2024), show promise in addressing this issue. Preliminary results indicate that these models perform better at longer lead times, and have a rate of performance decline more similar to IFS ENS than to

deterministic data-driven models (Price et al., 2024).

IFS HRES appears also to be overall best in terms of tail calibration (Figures 9– 12), even though differences between IFS HRES and data-driven models are small for forecasts of global extremes, especially at shorter lead times (Figure 9). Differences between the two data-driven models are also overall small, with GraphCast oftentimes performing better in the Tropics, and Pangu-Weather in the midlatitudes (Figures 4, 10–12). Additionally, Pangu-Weather appears to be better for cold extremes, and GraphCast for hot extremes.

In the main text, we compare the semi-operational versions of the data-driven models taking as input IFS HRES at time 0, with IFS HRES, using ERA 5 as ground truth for all the models. In Appendix D, we shift the focus to comparing reanalysis-based data-driven models and IFS HRES, utilizing different ground truths: ERA 5 for the data-driven models and IFS HRES at time 0 for the physics-based model. The findings in Appendix D generally support those in the main text (Figures D1-D6). As in the main, the data-driven models show an improvement over the physics-based model in terms of average skill, except for short-term 2m temperature forecasts (Figures D1 and D4), Additionally, the data-driven models are competitive in forecasting extreme events (Figures D2–D4), with a few exceptions. Specifically, IFS HRES continues to outperform all data-driven models in forecasting cold spells at short lead times, though this might partially be a consequence of the different ground-truths used for IFS HRES and the data-driven models. At the grid-point level (Figures D5–D12), the data-driven models are highly competitive in terms of average skill (Figure D5), with FuXi standing out for its remarkable performance in forecasting 2m temperature at longer lead times. In terms of extremes (Figure D6), IFS HRES remains superior over land at shorter lead times, but data-driven models progressively close the gap in the medium-range (Figure D12).

As suggested by previous literature, some additional challenges need to be addressed before data-driven models may be fully implemented operationally, including the lack of uncertainty information provided by the deterministic forecasts (Molina et al., 2023; de Burgh-Day and Leeuwenburg, 2023; Scher and Messori, 2021; Clare et al., 2021) and the lack of physical constraints in the forecasts generated by the models (Kashinath et al., 2021; Beucler et al., 2020). Moreover, with the exception of GraphCast, none of the data-driven models that we analysed here forecasts precipitation, which, when extreme, is a key meteorological hazard. Finally, further evaluations of extreme forecast behaviour may be necessary. Our analysis is limited to a narrow range of near-surface extremes and, due to current data availability, to extremes occurring in 2020. This limits our ability to draw conclusions on long-term performance. The short time period considered also exposes our results to sensitivity to low-frequency modes of climate variability, which modulate the occurrence of extreme events and may also affect their predictability (Goddard and Gershunov, 2020; Luo and Lau, 2020; Chartrand and Pausata, 2020). We therefore encourage more comprehensive evaluations in the near-future, as more data become available, and deep-learning models are extended to produce forecasts of other relevant variables for weather extremes (e.g. wind gusts and precipitation).

We also note that all forecast evaluation metrics, including those used here, suffer from limitations: For criteria 1 and 2 the RMSE is also the objective function of the ML models, which means that evaluating against RMSE is not a fully independent target. Additionally, criteria 1 and 2 are not proper or consistent scores, meaning that it would be possible to design a data-driven model optimising for tail RMSE and outperforming all other models while ignoring other aspects of performance (Taggart, 2022; Lerch et al., 2017). We note, though, that this limitation applies to most metrics of tail performance for deterministic models used in the previous literature, including, widely popular precision-recall curves and AUC ROC. Similarly, criterion 3 is only a measure of tail calibration, which can be maximised by post-processing schemes placing greater emphasis on tail behaviour than on the rest of the distribution. Because of this reason, inference based on any of those measure alone is not meaningful, and any tail comparison between models should be integrated by comparisons for the whole distribution of the variables, such as those presented in Figure 1 and Figure 5, and more qualitative measures of performance. Additionally, as highlighted by Watson (2022), raw measures of performance and qq-plots should also be complemented by a careful study of weather charts of case studies. In particular, we emphasise that better performance in just one of the three criteria used in this paper should not be interpreted in isolation as overall superiority of a model against the others.

To strengthen our results, we include, in Appendix B and C, two additional metrics of tail-performance which cannot be hedged by data-driven models in the same way as criteria 1 and 2 can. The results there are mostly in line with what shown in the main, suggesting that none of the data-driven models included in this paper has been hedging the tail-RMSE metrics included in our main analysis. However, the metrics presented in these appendices suffer from the fundamental limitation of selecting a part of or their whole extreme sample based on forecasts rather than on an independent ground-truth. This leads to a progressive deterioration of the selection criteria for the extremes with lead time, thus introducing a fundamental issue with the validity of the sample. Moreover, the fact that the sample becomes increasingly less representative of the ground-truth extremes at longer lead times, tends to favour data-driven models, which, as shown by the WeatherBench 2 (Rasp et al., 2024) and in this paper, are mostly superior in terms of standard metrics based on the overall distribution of near-surface variables.

We conclude that data-driven models can already compete with physics-based models in the forecast of near-surface temperature and wind extremes, but the performance of data-driven models varies by region, type of extreme event, and forecast lead time. The main challenges holding data-driven models back appear to be blurring, poor performance at high latitudes and lack of some key input variables. As solutions to blurring appear to be in sight, we argue that more attention should be given to loss functions and input variables. We would therefore encourage more studies in this direction, especially to investigate whether dropping latitude-based weights in the training routine might lead to better performance on extremes at higher latitudes.

As of now, we can already envisage a hybrid use of physics- and data-driven models to forecast extremes, with physics-based models being supplemented by data-driven models for those areas where data-driven models have been shown to be superior in terms of tail performance, such as in the tropics. This hybrid usage could take the form of a fully hybrid model,

such as the recent neural GCM (Kochkov et al., 2024), or even simple post-processing schemes based on weighted averages of physics-based and data-driven forecasts.

*Code and data availability.* The forecasts generated by all models are freely available through the WeatherBench 2 (Rasp et al., 2024). All the data-driven models are trained using the ERA 5 reanalysis dataset (Hersbach et al., 2020), which is freely available through the Copernicus Climate Change Service at https://doi.org/10.24381/cds.adbb2d47 and https://doi.org/10.24381/cds.bd0915c6, as well as through the WeatherBench 2 (Rasp et al., 2024). The code used to train the data-driven models included in the comparison are provided by the authors of the models themselves, and details an how to access the code and pre-trained models are provided in the respective papers (Bi et al., 2023; Lam et al., 2023; Chen et al., 2023b). The code developed by the authors of this paper to perform the comparisons and generate the plots included here is available on Zenodo at https://zenodo.org/records/13329880 (Olivetti, 2024), as well as on the Github page of the corresponding author, Leonardo Olivetti.

*Author contributions.* The authors are jointly responsible for the conceptualisation of this work including the visualisations, and all the revision and editing of the submitted manuscript. L. Olivetti has developed the code used for the model comparisons and to generate the visualisations, and written most of the original draft. G.Messori has acquired the funding and other resources necessary to conduct this research and provided extensive supervision.

*Competing interests.* The authors declare no conflicts of interest relevant to this study.

**Appendix A: Pixel by pixel comparisons including individual data-driven models**

This section includes complementary figures to figures 5-8, displaying which of the models is best at each pixel (Figure A1 and A2) and the magnitude of those differences (Figure A3 and A4).

# RMSE pixel by pixel - which model is best?

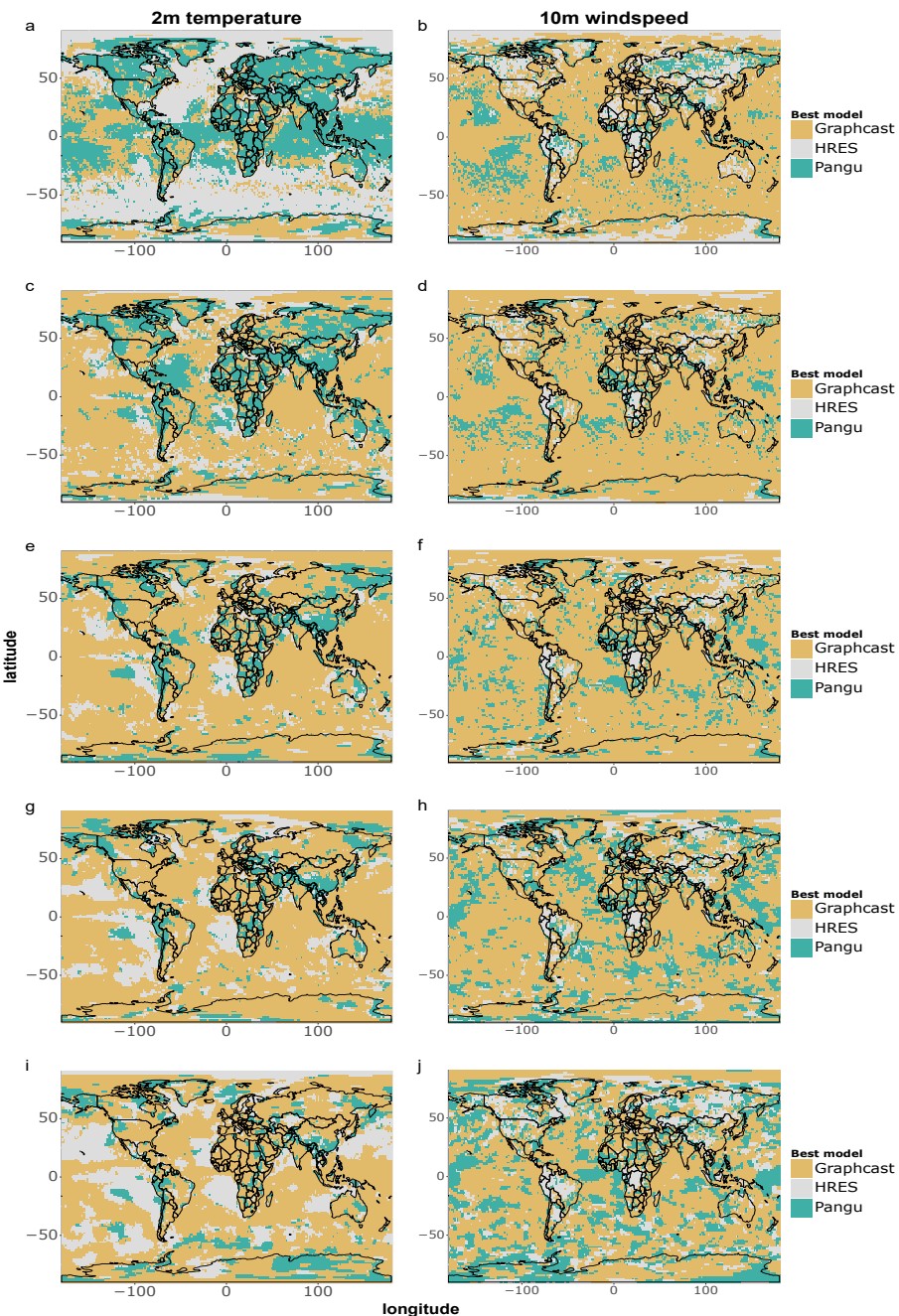

**Figure A1.** Single-grid point RMSE comparison for all data-points of 2m temperature and 10m windspeed. a-b) 1 day forecasts; c-d) 3 days forecasts; e-f) 5 days forecasts; g-h) 7 days forecasts; i-j) 10 days forecasts.

**RMSE pixel by pixel - which model is best?**

**Figure A2.** Single-grid point RMSE comparison for cold, hot and windspeed extremes. a-c) 1 day forecasts; d-f) 3 days forecasts; g-i) 5 days forecasts; j-l) 7 days forecasts; m-o) 10 days forecasts.

**RMSE pixel by pixel - magnitude of differences**

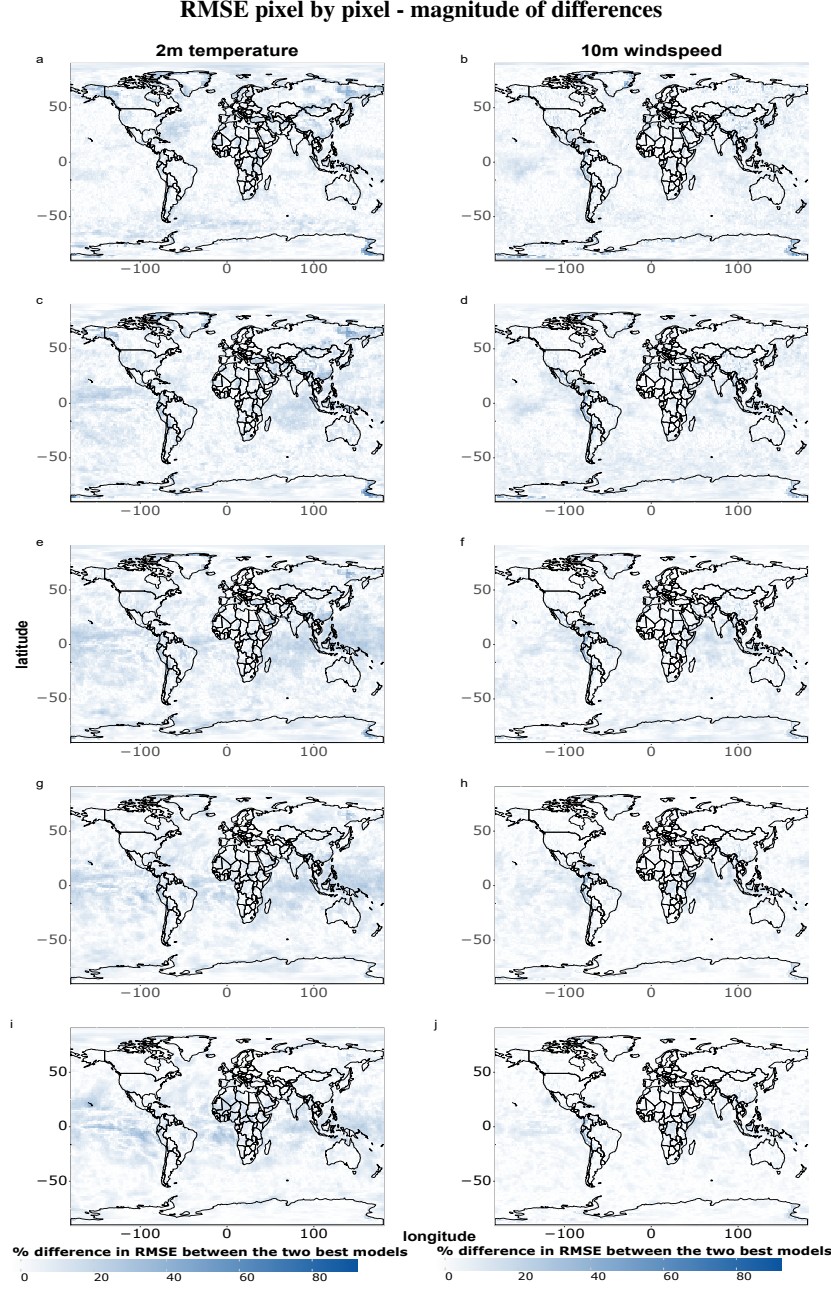

**Figure A3.** Magnitude of single-grid point RMSE differences between the two best models at that each grid point for all data-points of 2m temperature and 10m windspeed. a-c) 1 day forecasts; d-f) 3 days forecasts; g-i) 5 days forecasts; j-l) 7 days forecasts; m-o) 10 days forecasts.

# RMSE pixel by pixel - magnitude of differences

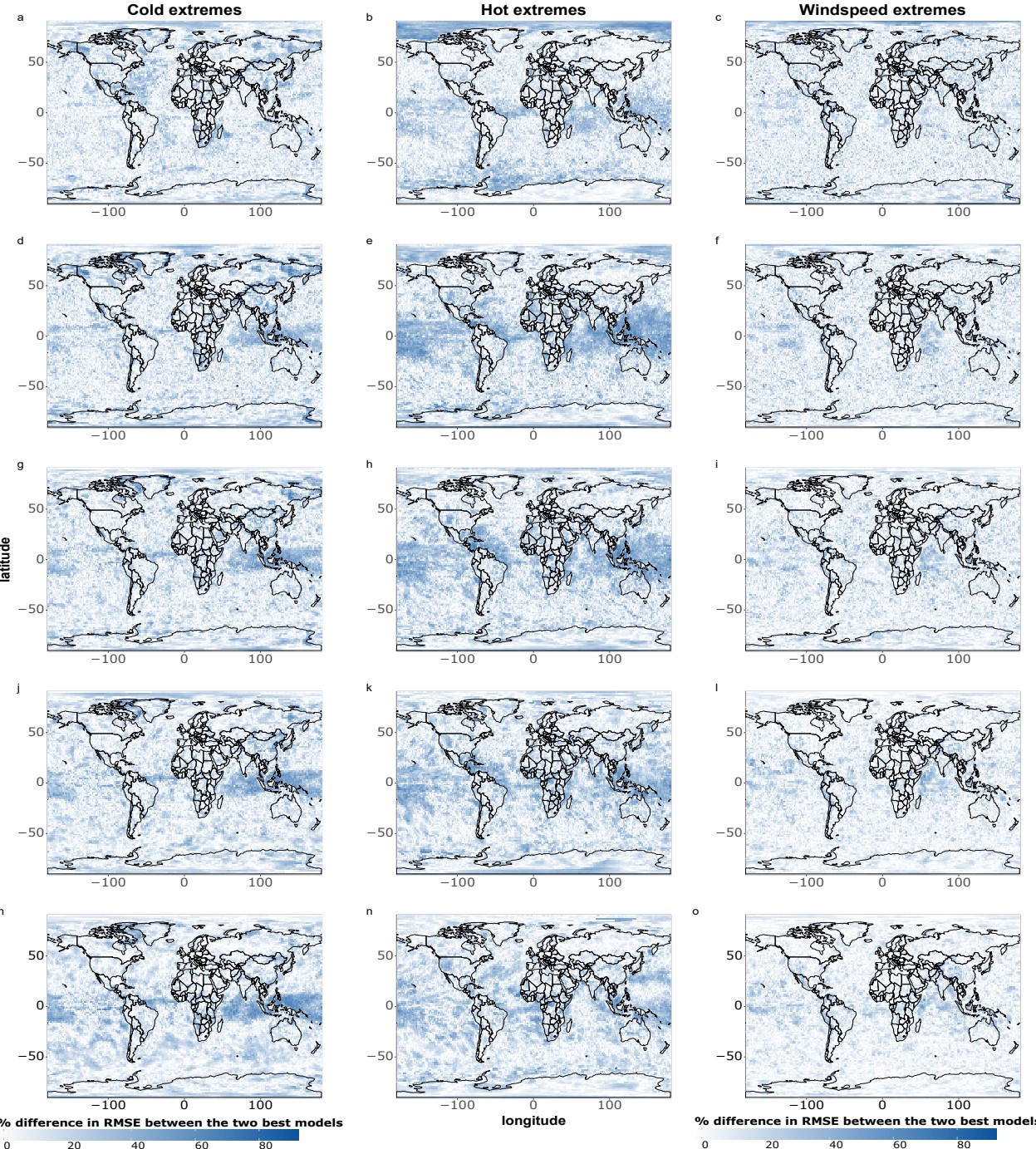

**Figure A4.** As Figure A3 but for hot cold, hot and windspeed extremes.

## Appendix B:  Comparison on consistent score emphasising tail performance

We report here the results of additional evaluations of tail performance based on Taggart (2022)'s MSE decomposition (Equation 1 and 2 in Taggart (2022)), where we emphasise performance in the tails by means of a rectangular partition, where the cut-off values are given by extreme quantiles of all ground-truth (ERA 5) data-points for the given region. In Figures B1 and B2, below, we include only the scores for the part of the decomposition emphasising tail performance (S1 for cold extremes and S2 for hot and windspeed extremes). Since S is the MSE in this case, it can be easily computed by squaring the values reported in Figure 1. Since S =S1+S2, the remaining part of the decomposition not displayed here can be obtained by subtracting the results reported below from the MSE (S).

## Consistent scores scorecard - 5% cut-off

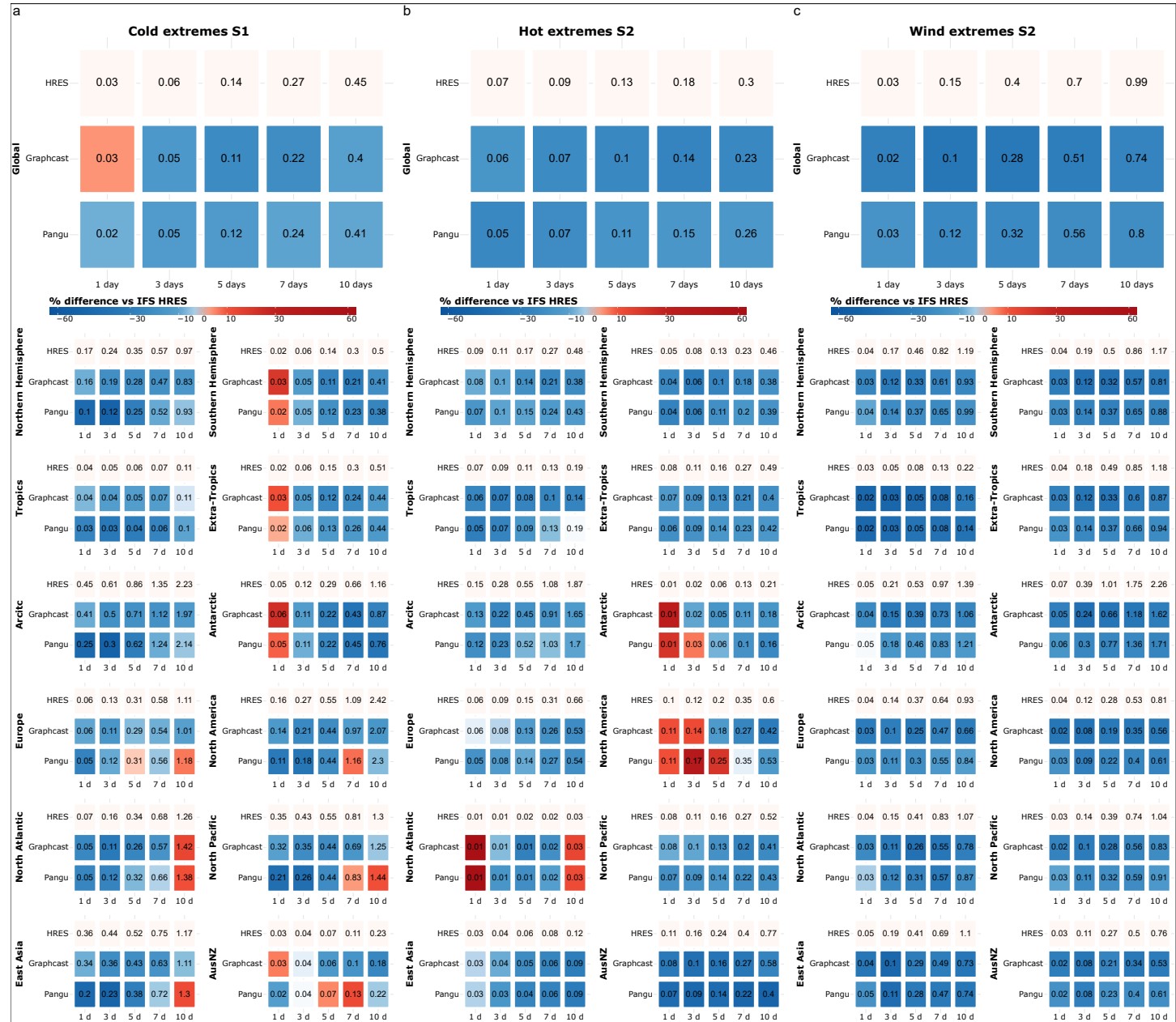

**Figure B1.** Scorecard for a) cold, b) hot and c) windy extremes based on rectangular partitions with a) 5th and b-c) 95th quantile of all test data-points in the given region as cut-off values. Blue shades indicate better performance than IFS HRES, red shades worse performance.

**Consistent scores scorecard - 1% cut-off**

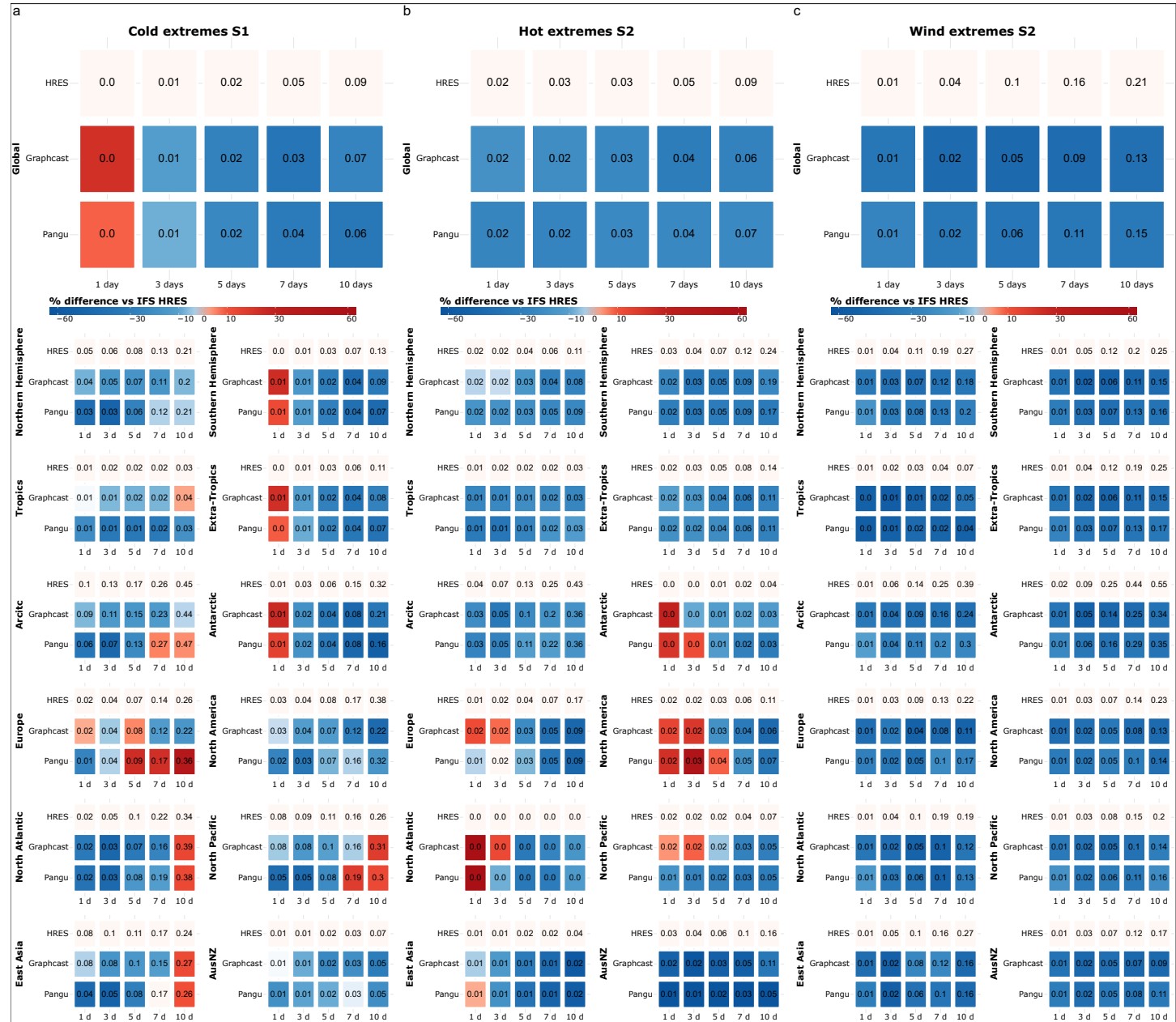

**Figure B2.** Scorecard for a) cold, b) hot and c) windy extremes based on rectangular partitions with a) 1th and b-c) 99th quantile of all test data-points in the given region as cut-off values. Blue shades indicate better performance than IFS HRES, red shades worse performance.

## Appendix C: Comparison on extremes selected based on the IFS HRES forecasts

We report here the results of additional evaluations of tail performance, where we select the extremes based on the IFS HRES forecast and respective quantile thresholds instead of the ground truth, i.e. the ERA 5 reanalysis. This approach has the advantage of preventing the risk of hedging by data-driven models, but it has the fundamental disadvantage of introducing validity issues in the extreme sample, since the quality of the forecasts and, therefore, the quality of the selection of the extremes, decreases with the lead time.

# RMSE scorecard for 5% most extreme data-points

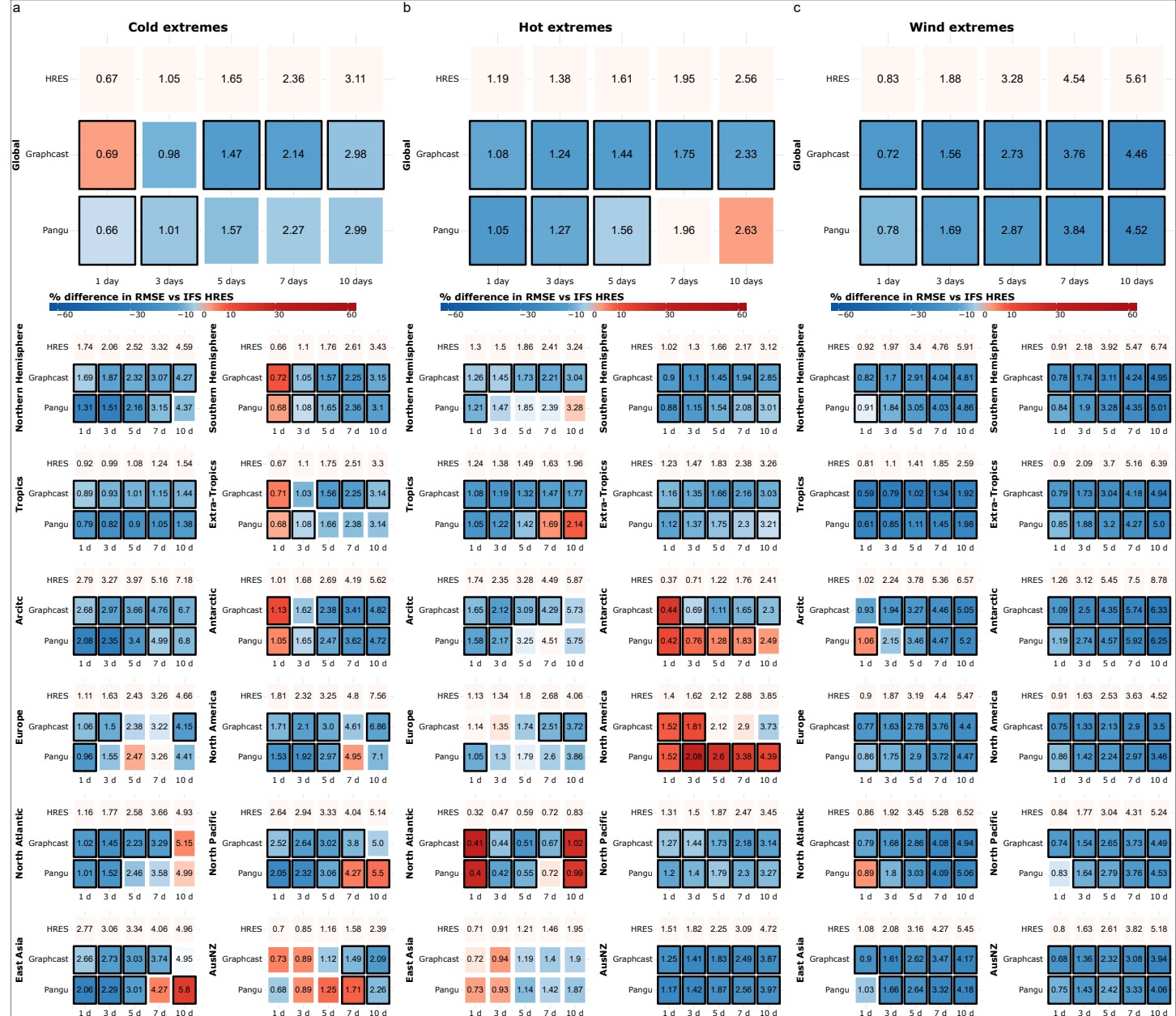

**Figure C1.** RMSE scorecard for cold (a), hot (b) and wind extremes (c) at a global and regional scale, computed on the (a) 5% lowest 2m temperature, (b) 5% highest 2m temperature and (c) 5% highest 10m windspeed data-points, respectively, selected on the basis of the IFS HRES forecast. Black borders indicate statistically significant differences in performance from IFS HRES, at the 5% level.

# RMSE scorecard for 1% most extreme data-points

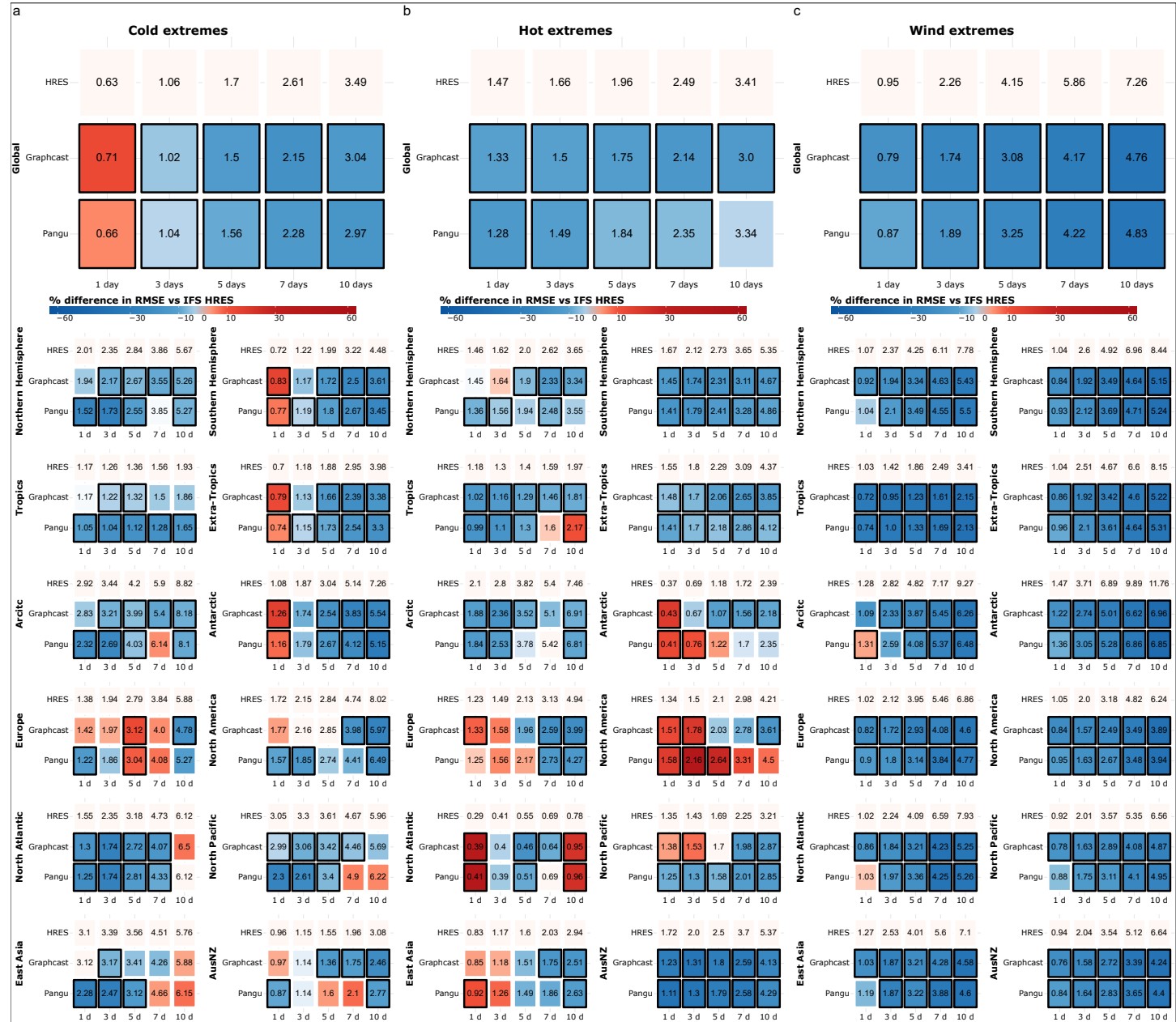

**Figure C2.** RMSE scorecard for cold (a), hot (b) and wind extremes (c) at a global and regional scale, computed on the (a) 1% lowest 2m temperature, (b) 1% highest 2m temperature and (c) 1% highest 10m windspeed data-points, respectively, selected on the basis of the IFS HRES forecast. Black borders indicate statistically significant differences in performance from IFS HRES, at the 5% level.

**Figure C3. a-f**: Best model in terms of tail-RMSE computed on the (a) 5% lowest 2m temperature, (b) 5% highest 2m temperature, (c) 5% highest 10m windspeed, (d) 1% lowest 2m temperature, (e) 1% highest 2m temperature, (f) 1% highest 10m windspeed data-points, selected on the basis of the IFS HRES forecast. **g-h** Best model in terms of overall RMSE for (g) 2m temperature and (h) 10m windspeed. Black borders indicate statistically significant better performance than the other models, at the 5% significance level.

**RMSE pixel by pixel - which model is best?**

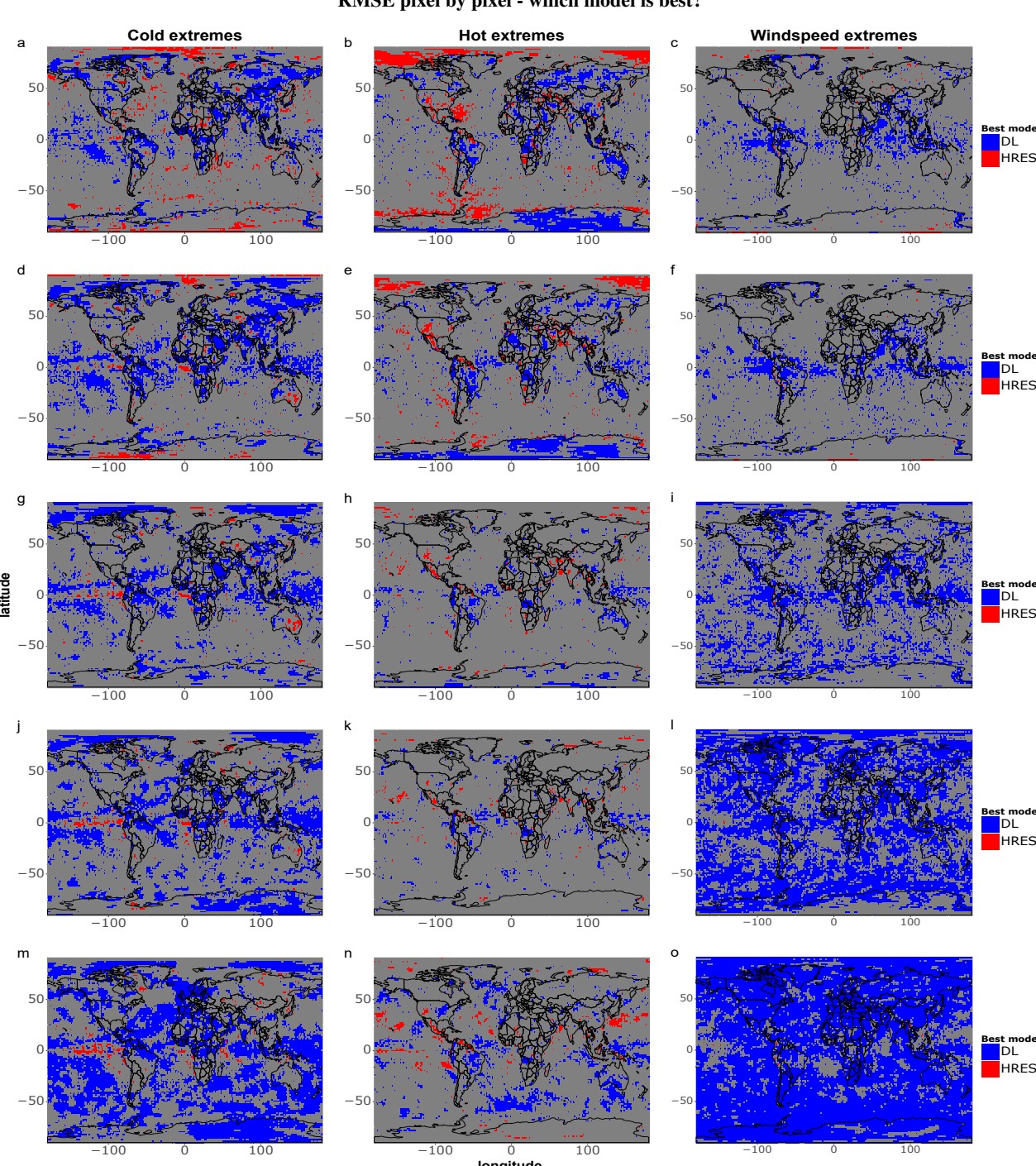

**Figure C4.** As in Figure 7, but selecting the extremes on the basis of the IFS HRES forecast instead of the ground-truth (ERA 5).

# RMSE pixel by pixel - magnitude of differences

**Figure C5.** As in Figure 8, but selecting the extremes on the basis of the IFS HRES forecast instead of the ground-truth (ERA 5)

## Appendix D:  Comparison of Reanalysis-based Data-driven Models

Here we provide global and regional scorecards and grid-point level comparisons for data-driven models using ERA5 reanalysis data as input. Following the WeatherBench 2 (Rasp et al., 2024), we attempt to make the comparison between reanalysis-based data driven models and IFS HRES as fair as possible by using IFS HRES t=0 as ground truth for IFS HRES, instead of ERA 5.

In this comparison, we also include FuXi (Chen et al., 2023b), a recent data-driven model building upon the work of Bi
et al. (2023). Forecasts generated by FuXi are currently available only for a reanalysis-based version of the model on the WeatherBench 2 (Rasp et al., 2024), which is why we include FuXi here but not in the comparisons in the main text. FuXi is trained on ERA5 reanalysis data for 1979 to 2017, and uses a vision transformer architecture (Dosovitskiy et al., 2020). FuXi's main innovation compared to previous models is its cascading optimisation approach, through which different sub-models are developed for different forecasting ranges, with the purpose of improving medium-to-long range forecasts (Chen et al., 2023b).

## RMSE scorecard based on all test data-points

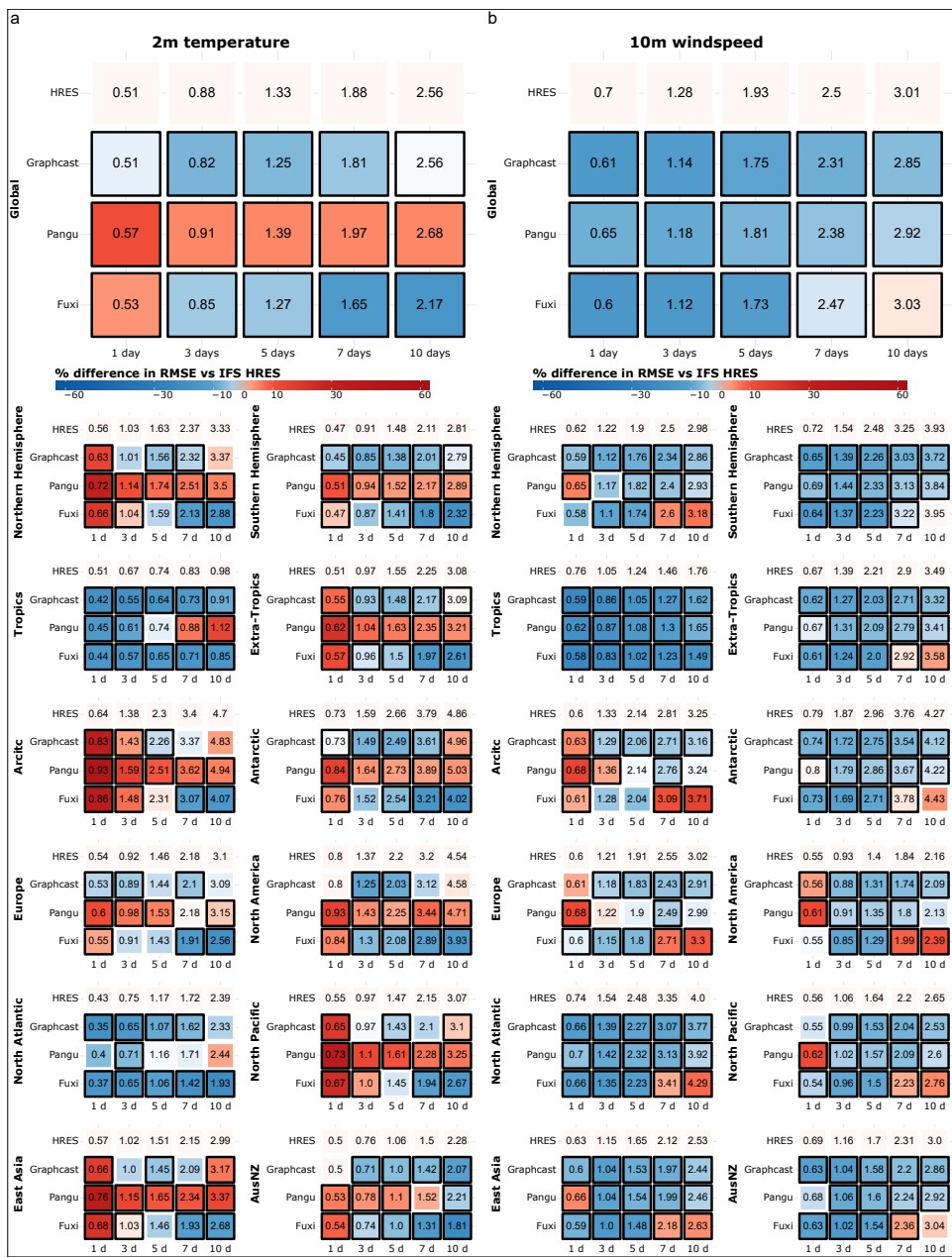

**Figure D1.** As in Figure 1, but using ERA5 as ground truth for the data-driven forecasts and IFS HRES at time 0 as ground-truth for the IFS HRES forecasts.

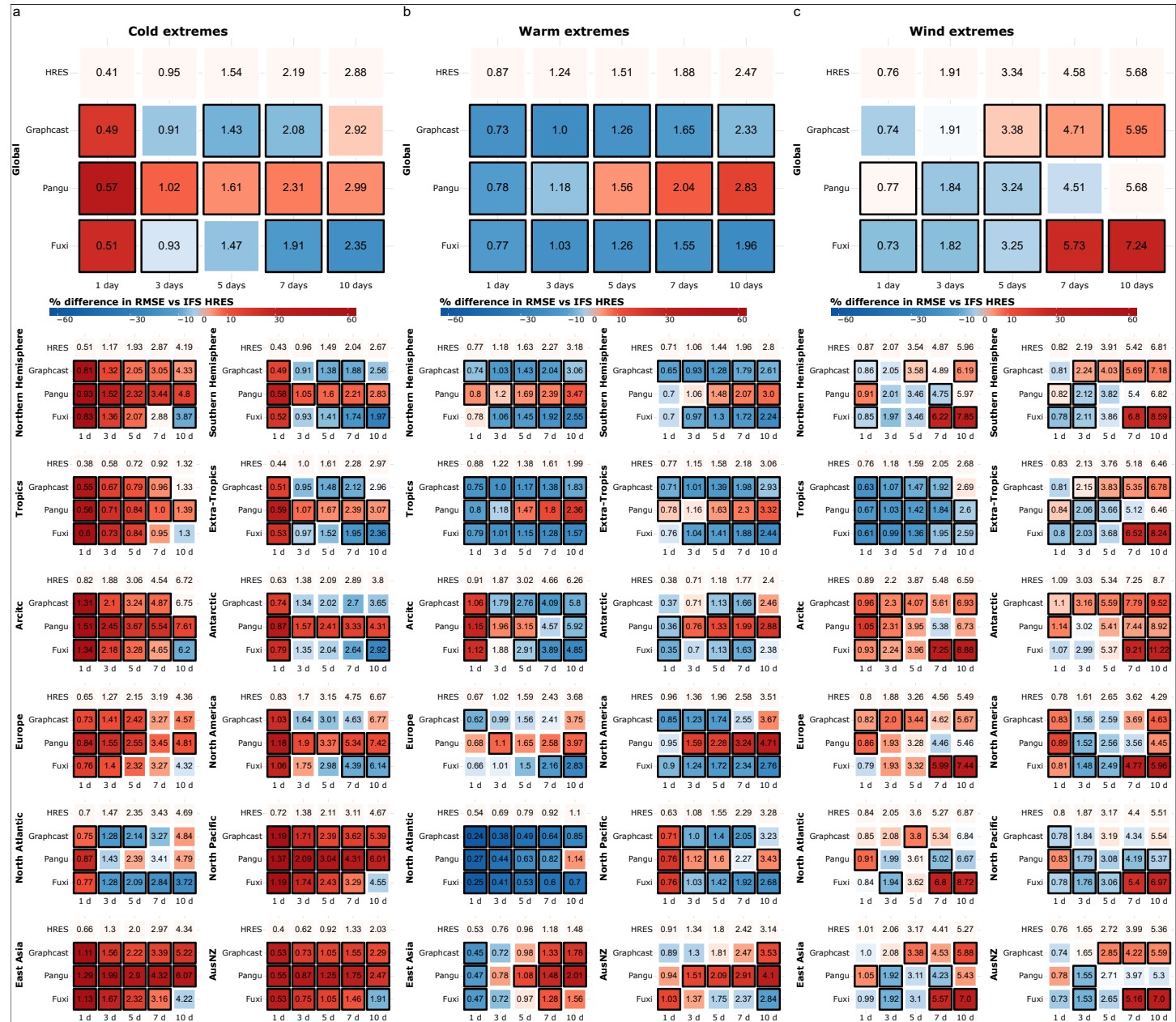

**Figure D2.** As in Figure 2, but using ERA5 as ground truth for the data-driven forecasts and IFS HRES at time 0 as ground-truth for the IFS HRES forecasts.

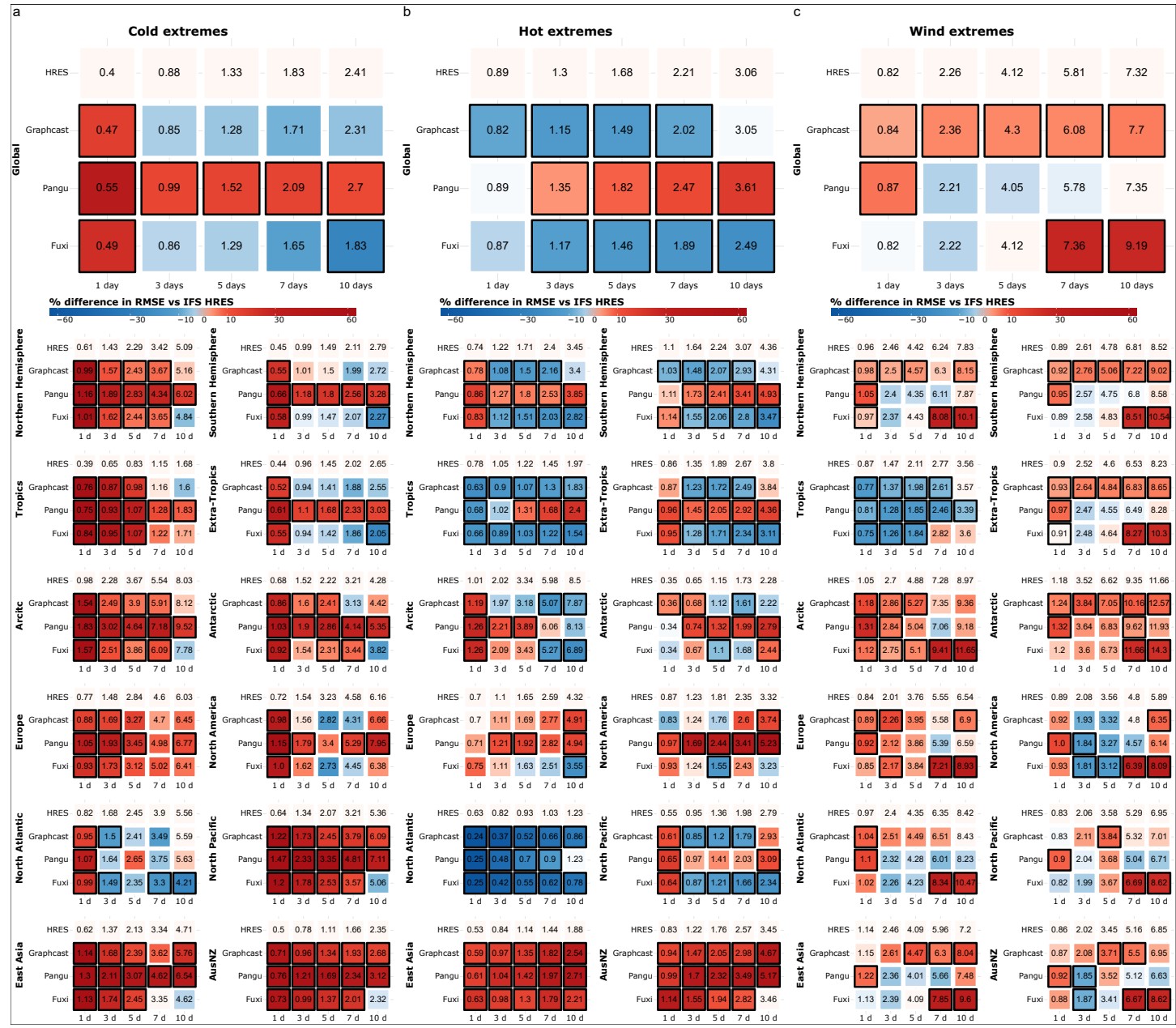

**Figure D3.** As in Figure 3, but using ERA5 as ground truth for the data-driven forecasts and IFS HRES at time 0 as ground-truth for the IFS HRES forecasts.

**Figure D4.** As in Figure 4, but using ERA5 as ground truth for the data-driven forecasts and IFS HRES at time 0 as ground-truth for the IFS HRES forecasts.

**Figure D5.** As in Figure 5, but including FuXi among the possible data-driven models and using ERA5 as ground truth for the data-driven forecasts and IFS HRES at time 0 as ground-truth for the IFS HRES forecasts.

**Figure D6.** As in Figure 7, but including FuXi among the possible data-driven models and using ERA5 as ground truth for the data-driven forecasts and IFS HRES at time 0 as ground-truth for the IFS HRES forecasts.

# RMSE pixel by pixel - magnitude of differences

**Figure D7.** As in Figure 6, but including FuXi among the possible data-driven models and using ERA5 as ground truth for the data-driven forecasts and IFS HRES at time 0 as ground-truth for the IFS HRES forecasts.

# RMSE pixel by pixel - magnitude of differences

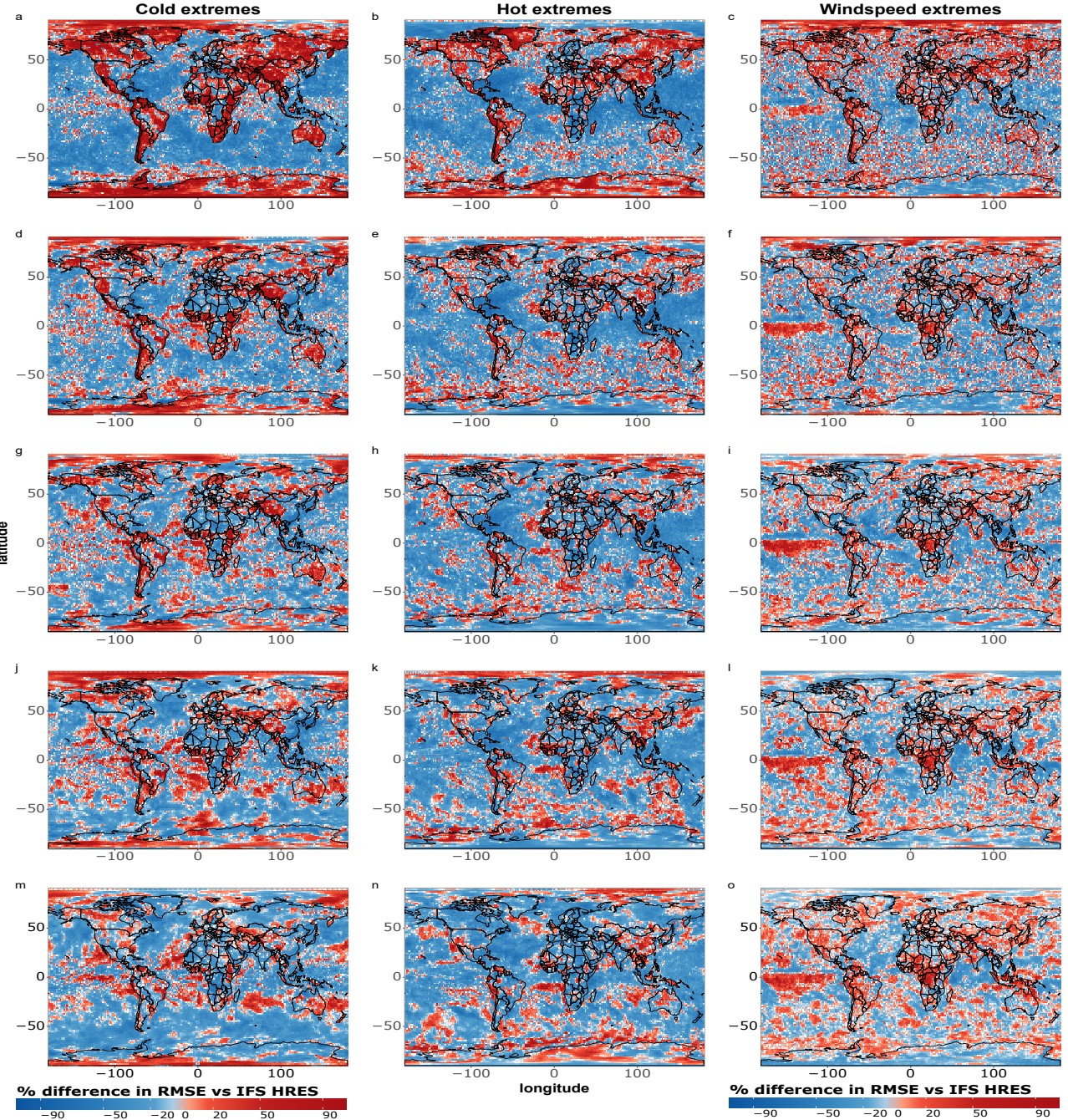

**Figure D8.** As in Figure 8, but including FuXi among the possible data-driven models and using ERA5 as ground truth for the data-driven forecasts and IFS HRES at time 0 as ground-truth for the IFS HRES forecasts.

## RMSE pixel by pixel - which model is best?

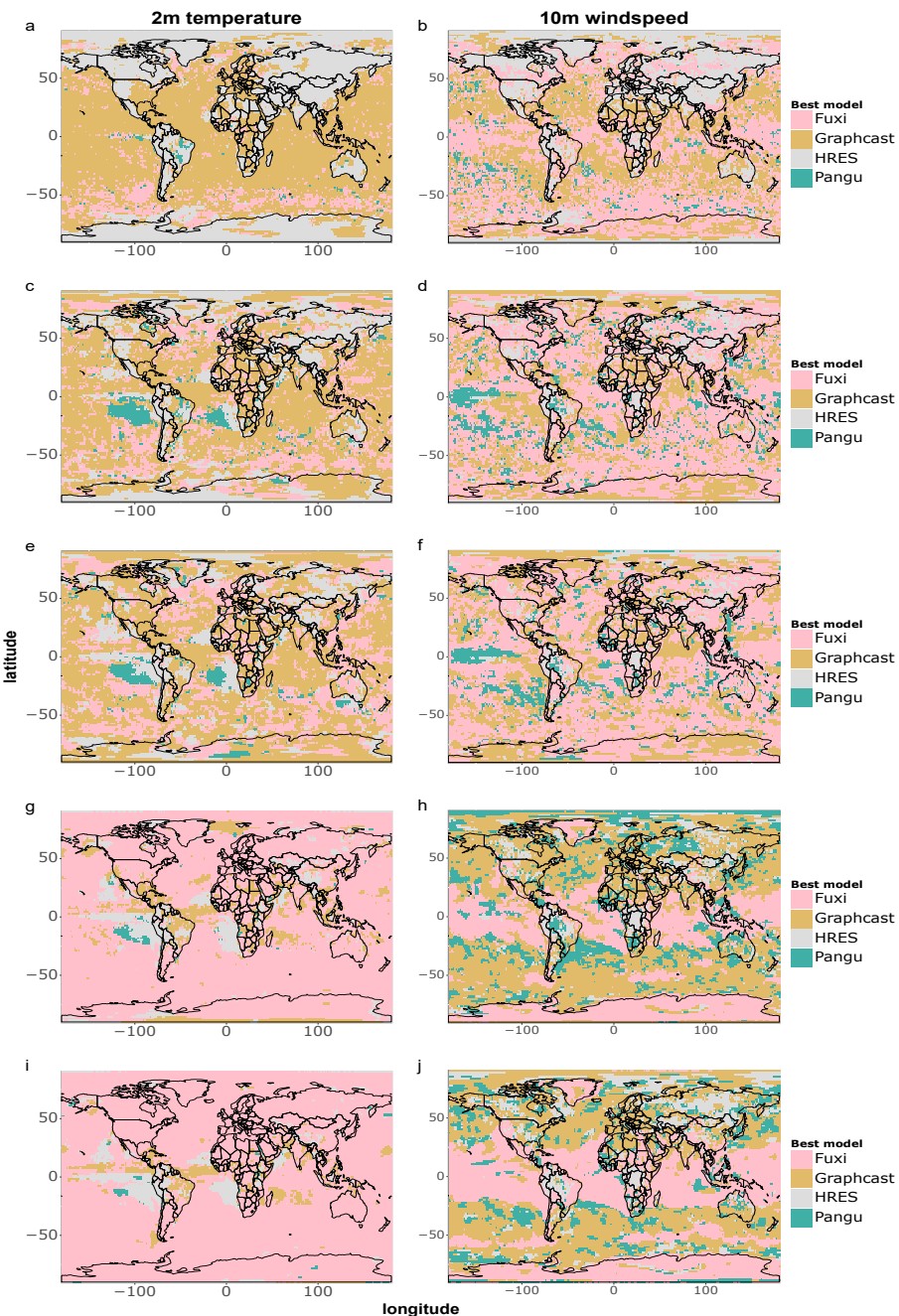

**Figure D9.** As in Figure A1, but using ERA5 as ground truth for the data-driven forecasts and IFS HRES at time 0 as ground-truth for the IFS HRES forecasts.

# RMSE pixel by pixel - which model is best?

**Figure D10.** As in Figure A2, but using ERA5 as ground truth for the data-driven forecasts and IFS HRES at time 0 as ground-truth for the IFS HRES forecasts.

**RMSE pixel by pixel - magnitude of differences**

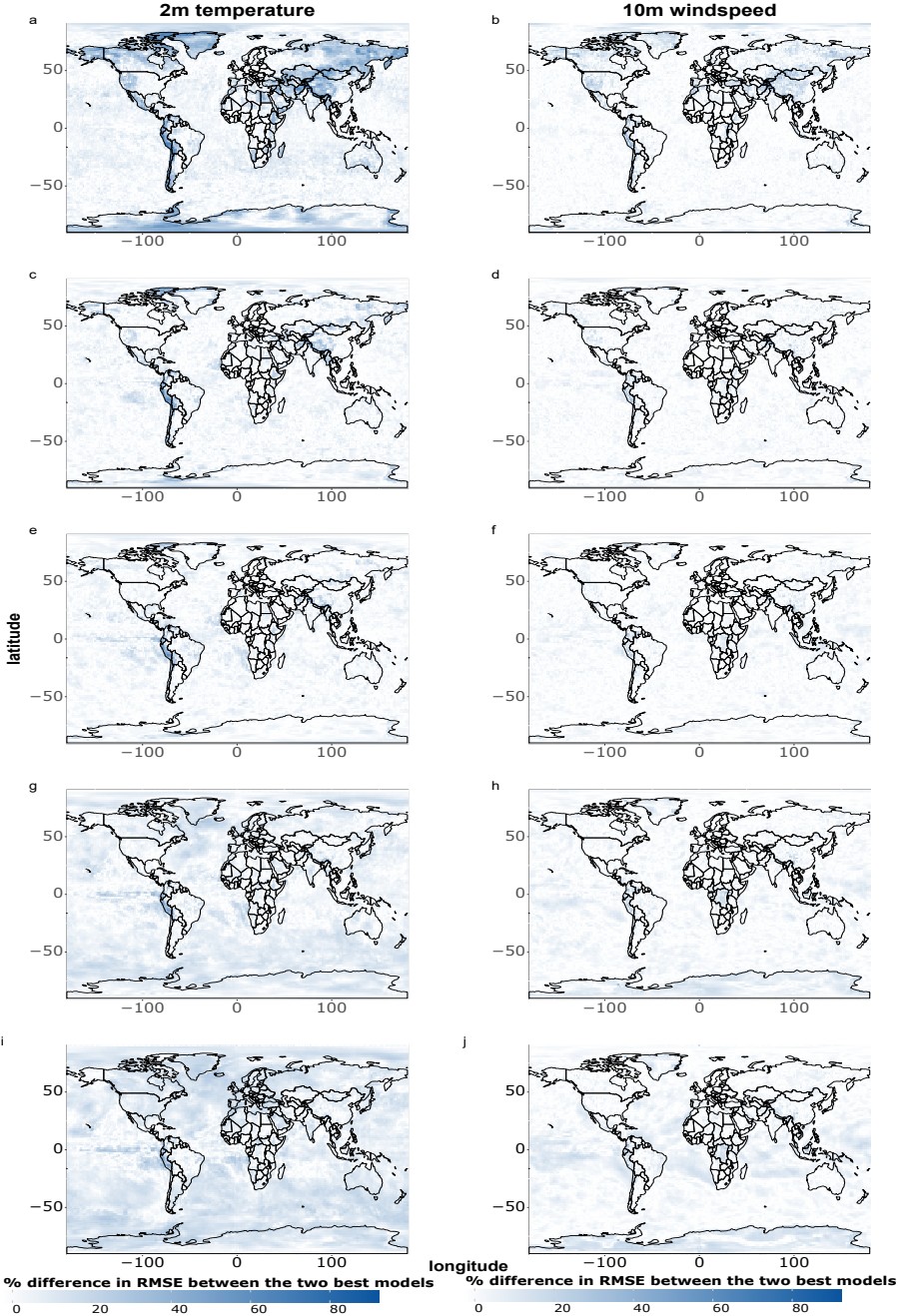

**Figure D11.** As in Figure A3, but using ERA5 as ground truth for the data-driven forecasts and IFS HRES at time 0 as ground-truth for the IFS HRES forecasts.

**RMSE pixel by pixel - magnitude of differences**

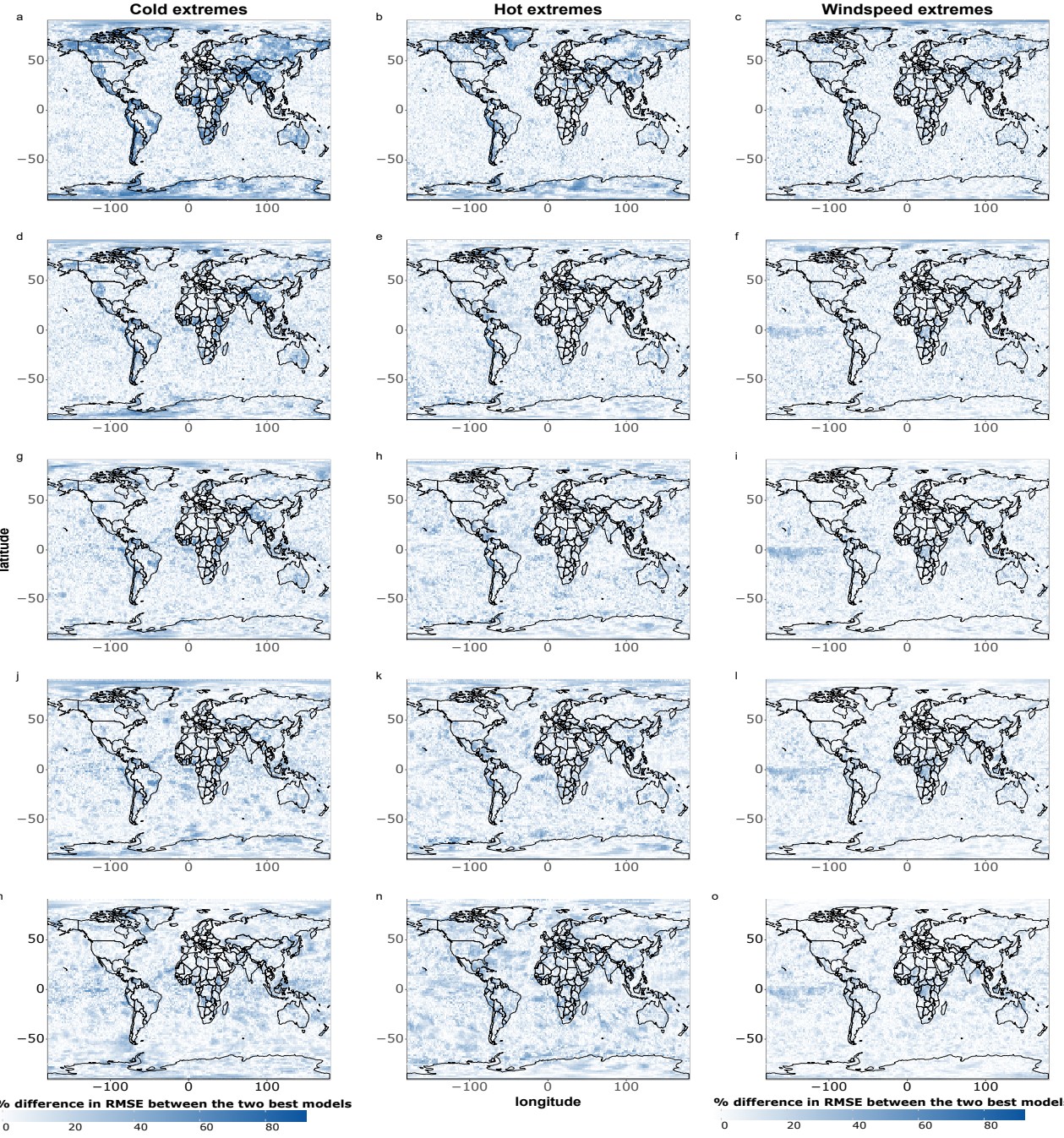

**Figure D12.** As in Figure A4, but using ERA5 as ground truth for the data-driven forecasts and IFS HRES at time 0 as ground-truth for the IFS HRES forecasts.

*Acknowledgements.* The authors thankfully acknowledge the support of the European Research Council (ERC) under the European Union's Horizon 2020 research and innovation programme (project CENAE: compound Climate Extremes in North America and Europe: from dynamics to predictability, Grant Agreement No. 948309). The computations and storage were aided by resources in project NAISS 2023/22-1356B and NAISS 2023/23-665, provided by the National Academic Infrastructure for Supercomputing in Sweden (NAISS) at C3SE, partially funded by the Swedish Research Council through grant agreement no. 2022-06725. The authors also acknowledge valuable discus-
sions with M. Krouma and S. Lerch, and are grateful for the valuable feedback provided by two anonymous reviewers, which contributed to improving the quality of this paper.

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
