# Peer review of "Do data-driven models beat numerical models in forecasting weather extremes? A comparison of IFS HRES, Pangu-Weather and GraphCast"

_EGUsphere, 2024_

## Referee Comment (RC1)

In their paper "Do data-driven models beat numerical models in forecasting weather extremes? A comparison of IFS HRES, Pangu-Weather and GraphCast", the authors provide an evaluation of the mentioned models plus FuXi for the whole distribution, and for the tails of the distribution (at 5% and 1% levels). It's really good to see this kind of analysis, as the evaluation of the new generation of data-driven models beyond simple things like global mean RMSE or individual case studies is very limited at present. I really enjoyed reading this paper, and am grateful to the authors for adding this valuable analysis to the body of knowledge on data-driven weather and climate models.

The authors do a good job of describing the data sets used, and their methodology. However, I feel like some additional figures will help to better contextualize their results – specifically where they present maps showing which model performs best at each grid point and lead time. Without corresponding maps showing the magnitude of the difference in scores between each model, it is hard to know how significant the patterns of which model is best are.

My main concern with this paper is that there is no discussion of the statistical significance of their results, which I think is essential given that they are looking at the tails of the distribution, and in some instances some quite small samples. I suggest the authors quote statistical significance thresholds of all their results and add stippling to their figures to indicate statistically significant areas.

More specific comments follow.

Abstract

Line 5-6: "in the average prediction of 10m windspeed and 2m temperature" - averaged over what? At what leadtimes and timescales?

Introduction

Line 49: "comparison in terms of a standard metric (RMSE)" - it would be good to note that RMSE is the objective function of the ML models, so evaluating against RMSE is not a fully independent target (I know you touched on this above – but making it clear you're taking this into consideration when you choose RMSE for your evaluation metric would be good).

Models and Methodology

Line 71-72: "operational Pangu weather (Bi et al., 2023) and operational GraphCast (Lam et al., 2023)" - I think you need to explain the difference between operational GraphCast/Pangu weather and reanalysis versions. I assume it's that the operational versions have been fine-tuned on the IFS HRES, but you should make that clear, especially since in the model description sections 2.2 and 2.3 you describe Pangu and GraphCast as being trained on ERA5 and predicting on the ERA5 grid, but then also talk about them predicting on the IFS HRES grid without an explanation of how that works. This is also important to be clear about, since (for example) providing IFS HRES ICs to a model trained on ERA5 then fine-tuned on IFS HRES will give a much better-quantified result than providing some other ICs to the model that it

has not been fine-tuned on – in this case some reduction in performance can be expected (based on my own investigations) but the extent of this would be unknown.

Line 76-79: I'm a little disappointed that you didn't also include an SFNO-based model like FourCastNet, since they exhibit quite different spatial variation in their RMSE scores compared to GraphCast, and are quite a different architectural approach to any of the models you've looked at.

Line 109: "As for Pangu-Weather" - suggest changing to "As with Pangu-Weather"

Section 2.4: Given you only look at FuXi in the Appendix, I'd suggest moving this model description there.

Line 122: "and all comparisons are based on a spatial resolution of 1.5 degrees" - I think it would be good to expand on this a bit. I assume you have regridded the data from 0.1 to 1.5 degree resolution - How did you do this? Any special treatment of the poles? Did you do this before or after calculating metrics (presumably before). Basically, some more procedural details would be appreciated.

Line 123: "sand" -> "and"

Line 126-139: I suggest putting this information in a table for easier reading.

Line 154 and onwards: I would suggest avoiding use of the term 'observations' since you are evaluating against reanalysis. The term observations runs the risk of causing confusion and giving the impression you are evaluating directly against point obs for example.

Line 169: So in case 2, the quantiles are computed using 702 values because that's what there is for each grid point? Might be worth making this clear, and making it clear that the number of points contributing in case 1 is 702 * num_lat * num_lon

Results

Fig 1: It might make the figure a bit too busy, but have you tried drawing borders around the scores? As it is the HRES scores look a bit like headings because their colour is always white. Borders may not look better though – it's hard to tell without seeing it, so please take this as just a loose suggestion.

Line 193: "extremes observations" -> "extreme observations"

Line 193: How many data points does the 5% most extreme cases leave you with? What is the statistical significance of the scores shown in Figure 2 (and Figure 1 as well for completeness I suppose).

Line 200: Ditto previous comment, but for 1%

Figure 4: I would suggest you try color schemes other than red-white-blue for this figure since there's a value judgement (better/worse compared to baseline) associated with those colours from the previous figures. Since this is a straight model comparison, some totally different color scheme would be good in my opinion.

Figure 5: Same comment as for Fig 4

Figure 5: EDIT – never mind – I see that you have addressed this in the discussion!

Figure 5: Similarly to the previous comment, it looks to me like there's some tendency for HRES to be better at 2T on the westward side of the continents, and worse on the eastward side. Do you have any thoughts on what this could be due to? Since these are upwelling regions and the feature grows with lead time my intuition is that it could be related to the lack of an ocean in the ML models?

Figures 5 and 6: Some measure of the magnitude of the differences between the models would be very valuable to contextualize what's shown in this figure (maybe maps of the model differences with stippling for statistical significance added to another appendix?) – some of these pixels where one model is shown as best may have a very marginal difference between the models and it would be good to know where this is the case. It would also be good to include an indication of what is statistically significant with this, especially for Fig 6 (where I think your sample is 702 * 0.05 = 35?). This could for example be stippling on Figs 5 and 6 as well as any maps of the magnitudes of the differences.

Figure 7: I find these plots pretty hard to read without leaning right in – perhaps you could increase the marker size, or subtract the y=x line from each set or points and display them ad deviations from perfect calibration to increase the visual distance between the markers?

Figure 7: Some indication of statistical significance or confidence on these plots would be good

Figure 8: Ditto both comments from Fig 7

Line 261-277: While this is interesting, it feels a little out of place since you haven't discussed FuXi anywhere else, and in the next section you once again stop referring to FuXi. I feel like you should either move these paras to the appendix so that the FuXi and other reanalysis initialized models are self-contained in the appendix, or you should add some acknowledgement of their analysis to the opening sentences of the conclusion to make the transition from these paragraphs to the conclusion less jarring.

Discussion and Conclusions

Line 295: It also looks to me like Pangu does better at cold extremes, and for hot extremes GraphCast is better over the oceans while Pangu is better over land (based on Fig 6)

Line 298-299: "the choice of best model depends strongly on region, lead time, type of extreme and in some cases even level of extremeness". I feel like this might not be quite the right way to put it – to me this implies that there is strong variability in the magnitude of the differences between the model's scores with region, lead, extreme type etc., but the magnitude of the differences between the models is not clear form most of your figures. All we know is that there is a lot of variability in which is best with region, lead time etc., but not by how much it is best. I suspect that you meant that with your wording, but I'm a bit worried it could be misinterpreted and suggest you revise this statement.

Line 301-303: ". Ideally, we envisage a hybrid use of physics- and data-driven models to forecast extremes, with physics-based models being supplemented by data-driven models for those areas where

data-driven models have been shown to be superior in terms of tail performance." - are the sizes of the differences in performance enough to justify this approach? I'm taking an operational forecast perspective here, and it feels like the potential gains would have to be more than marginal to justify this.

Appendix

The same comments apply to the appendix figures as to the figures in the main text:

- Some measure of statistical significance would be useful
- For figures A5 and A6, some accompanying figures showing the magnitude of the differences between the models would help give perspective on the significance of the spatial patterns in the figure.
- For figures A4, A5 and A6, a different colour scheme would work better I think – one where each of the models is given a different colour (not just shades of the same colour), and a scheme which is not the same as the one used for showing the magnitude of the differences in scores (since this has a value judgement attached to it)..

---

## Author Comment (AC1)

**Answer to Review #1**

Dear Reviewer,

Thank you very much for your positive remarks and constructive feedback! We really appreciate the time you have spent reviewing our work. Below, you find a line by line answer to your comments:

**Abstract**

Line 5-6: "in the average prediction of 10m windspeed and 2m temperature" - averaged over what? At what leadtimes and timescales?

We will change this sentence to "We find that data-driven models outperform ECMWF's physics-based deterministic model in terms of global RMSE for 1d-10d ahead forecasts, and can also compete in terms of extreme weather predictions in most regions. "

Introduction Line 49: "comparison in terms of a standard metric (RMSE)" - it would be good to note that RMSE is the objective function of the ML models, so evaluating against RMSE is not a fully independent target (I know you touched on this above – but making it clear you're taking this into consideration when you choose RMSE for your evaluation metric would be good).

Thank you for pointing this out, we agree with the reviewer that it is important to emphasise more this point in the manuscript. We will add a paragraph to the discussion section explaining the limitations of the extreme metrics approach (including the limitation pointed out by the reviewer), and inviting practitioners to integrate our analysis with case-studies and qualitative analyses of weather charts.

**Models and Methodology**

Line 71-72: "operational Pangu weather (Bi et al., 2023) and operational GraphCast (Lam et al., 2023)" - I think you need to explain the difference between operational GraphCast/Pangu weather and reanalysis versions. I assume it's that the operational versions have been fine-tuned on the IFS HRES, but you should make that clear, especially since in the model description sections 2.2 and 2.3 you describe Pangu and GraphCast as being trained on ERA5 and predicting on the ERA5 grid, but then also talk about them predicting on the IFS HRES grid without an explanation of how that works. This is also important to be clear about, since (for example) providing IFS HRES ICs to a model trained on ERA5 then fine-tuned on IFS HRES will give a much better-quantified result than providing some other ICs to the model that it has not been fine-tuned on – in this case some reduction in performance can be expected (based on my own investigations) but the extent of this would be unknown.

We agree that a concise explanation of what is meant by "operational version" is needed. In this context, "operational version" means that the model is able to generate forecasts based exclusively on inputs available within IFS HRES - without using reanalysis data or variables which are not part of the operational analysis (e.g. precipitation) to produce new forecasts.

Pangu operational is exactly the same model as the original Pangu-Weather, yet being fed IFS HRES instead of reanalysis data. GraphCast operational is a slightly modified version of the original model, which, differently from the original model, does not require precipitation as input to generate new forecasts. Furthermore, GraphCast operational has been undergoing a slight fine-tuning of the parameters to compensate for the loss of precipitation as input. In this respect, we agree that GraphCast has an edge over Pangu. However, it is up to the model developers to decide to what extent to fine-tune their models, and which versions of their models to make available to the public. As neither we nor the WeatherBench 2 can affect this process, we think it still makes sense to compare the best available versions of each model in our benchmark. Following the reviewer's comment, we will add  an explanation of the differences between operational and non-operational versions of GraphCast and Pangu-Weather to the respective sections introducing these models (Subsections 2.2-2.3).

Line 76-79: I'm a little disappointed that you didn't also include an SFNO-based model like FourCastNet, since they exhibit quite different spatial variation in their RMSE scores compared to GraphCast, and are quite a different architectural approach to any of the models you've looked at.

We agree that it would be interesting to include a SFNO-based model to the comparison. However, we are limited here by data availability since none of these models were included in the WeatherBench 2 (Rasp et al., 2024), and predictions produced by other tools (e.g. ECMWF plugin tool) are not comparable to the one included in the WeatherBench 2, as they do not take IFS HRES as input. It is our hope that some future work may include more models!

Line 109: "As for Pangu-Weather" - suggest changing to "As with Pangu-Weather"

We will make  this change to the text.

Section 2.4: Given you only look at FuXi in the Appendix, I'd suggest moving this model description there.

We agree with the reviewer and will move FuXi's model description to the new Appendix D.

Line 122: "and all comparisons are based on a spatial resolution of 1.5 degrees" - I think it would be good to expand on this a bit. I assume you have regridded the data from 0.1 to 1.5 degree resolution - How did you do this? Any special treatment of the poles? Did you do this before or after calculating metrics (presumably before). Basically, some more procedural details would be appreciated.

All data have been regridded from their original resolution to a 240x121 equiangular conservative grid. This was done by ECMWF for the ECMWF data, and by the WeatherBench 2 for all other models. More details can be found in the WeatherBench 2 paper (Rasp et al., 2024). We will add a brief explanation to the manuscript.

Line 123: "sand" -> "and"

Thank you for spotting this type, we will correct it in text.

Line 126-139: I suggest putting this information in a table for easier reading.

We will implement the reviewer's suggestion and create a new table (Table 1) to include this information.

Line 154 and onwards: I would suggest avoiding use of the term 'observations' since you are evaluating against reanalysis. The term observations runs the risk of causing confusion and giving the impression you are evaluating directly against point obs for example.

We will replace the term "observations" with "data-points".

Line 169: So in case 2, the quantiles are computed using 702 values because that's what there is for each grid point? Might be worth making this clear, and making it clear that the number of points contributing in case 1 is 702 * num_lat * num_lon

Yes, that is correct. We will add a sentence to each of the two criteria to clarify the number of available data-points for evaluation.

**Results**

Fig 1:
It might make the figure a bit too busy, but have you tried drawing borders around the scores? As it is the HRES scores look a bit like headings because their colour is always white. Borders may not look better though – it's hard to tell without seeing it, so please take this as just a loose suggestion.

We introduced the use of borders in Figure 1 to indicate statistical significance instead. See our reply to the in-depth comment below.

Line 193: "extremes observations" -> "extreme observations"

Thank you for pointing out this typo, we have implemented this change in the manuscript

Line 193: How many data points does the 5% most extreme cases leave you with? What is the statistical significance of the scores shown in Figure 2 (and Figure 1 as well for completeness I suppose).

We welcome this suggestion, as introducing some form of statistical significance could help to separate meaningful differences between models from overall noise. However, we should be mindful of the fact that in this case standard tests of significance may tend to overestimate the significance of the differences between models, since the selected extreme

observations may be strongly correlated in space and time. To address this issue, we make use here of a paired t-test with cluster robust standard errors, which takes into account the clustered nature of the selected observations in space and in time. This is an approach issuing from panel data econometrics, first introduced by Liang and Zeger (1986) and Arellano (1987) and recently revamped by in depth treatments by Angrist and Pischke (2009), and Cameron and Miller (2015). The intuition behind the use of clustered standard errors in this case is that since many of our data-points come from adjacent grid points and are clustered in space and time, the effective number of degrees of freedom of our test is much smaller than the number of available paired forecast differences. Thus, we correct for this by inflating the standard errors, introducing a clustering parameter.

Below we provide an example of the new RMSE scorecards figures, where black borders indicate that the performance of the model at a given lead time is statistically significantly different from IFS HRES, at the 5% significance level.

[Figure]

Line 200: Ditto previous comment, but for 1%

See answer to previous comment, we updated the figure for the 1% extremes as we did for the figure for 5% extremes.

Figure 4: I would suggest you try color schemes other than red-white-blue for this figure since there's a value judgement (better/worse compared to baseline) associated with those colours from the previous figures. Since this is a straight model comparison, some totally different color scheme would be good in my opinion. Figure 5: Same comment as for Fig 4.

We understand the reviewer's concern, and have changed the colour scheme accordingly, see example figure (Fig.4) below.

[Figure]

Figure 5: Similarly to the previous comment, it looks to me like there's some tendency for HRES to be better at 2T on the westward side of the continents, and worse on the eastward side. Do you have any thoughts on what this could be due to? Since these are upwelling regions and the feature grows with lead time my intuition is that it could be related to the lack of an ocean in the ML models?

This is certainly a possible explanation. Currently, there are interesting discussions underway within the weather ML community about the possibility of developing coupled ML

models that could tackle this challenge, which is especially significant at longer time scales. However, given the limited data availability and the fact that we have limited evidence to support this claim, we believe it may be best to express ourselves with caution on the topic in the manuscript. Nevertheless, we are going to briefly mention this point in our discussion.

Figures 5 and 6: Some measure of the magnitude of the differences between the models would be very valuable to contextualize what's shown in this figure (maybe maps of the model differences with stippling for statistical significance added to another appendix?) – some of these pixels where one model is shown as best may have a very marginal difference between the models and it would be good to know where this is the case. It would also be good to include an indication of what is statistically significant with this, especially for Fig 6 (where I think your sample is 702 * 0.05 = 35?). This could for example be stippling on Figs 5 and 6 as well as any maps of the magnitudes of the differences.

We agree with the concern raised by the reviewer, and we have added some figures showing statistically significant differences between the performance of machine learning models and IFS HRES (See new example figure below). The significance tests for these figures also use the aforementioned clustered standard errors to account for clustering of extremes in time, and global false discovering rates (Wilks 2016, Benjamini and Hochberg, 1995) to correct for multiple testing and ensure robust statistical inference. As illustrated by Wilks (2016), this approach is also robust for spatially correlated values. In the figure, dark grey areas show lack of statistical significance of the results, namely that the difference between IFS HRES and the best machine learning models is not statistically significant . Indeed, as suggested by the reviewer, many results are not statistically significant in Figure 6, due to the small sample size (n=36 at each grid point). However, some interesting regional patterns still emerge, which we discuss both in relation to this figure and in Section 4 in the manuscript.  As in Figures 1-3, blue shades indicate that the machine learning model is better than IFS HRES, whereas red shades indicate that IFS HRES is better.

[Figure]

To complement these new figures, as suggested by the reviewer, we also plan to add corresponding figures showing the magnitude of the differences between the models, see example figure below.

[Figure]

Figure 7: I find these plots pretty hard to read without leaning right in – perhaps you could increase the marker size, or subtract the y=x line from each set or points and display them ad deviations from perfect calibration to increase the visual distance between the markers?

We will increase the marker size as suggested by the reviewer and will also change the colour scheme to reflect changes in Figures 1-3.

Figure 7: Some indication of statistical significance or confidence on these plots would be good

Figures 7-10 are tail reliability plots (qq-plots for the tails of the distribution) that show the calibration of the forecast vs ERA 5. This type of plots does not usually come with confidence intervals or significance assessments attached, since they do not aim to rank the forecasts or show any significant difference between them, namely they do not aim to perform any kind of forecast-based inference. They just aim to provide a rough visual assessment of the calibration of the forecast at different quantiles during the test period vs the (imperfect) ground truth. Any significance assessment here would require making assumptions on the distribution of future extreme observations, on the correctness of the ground-truth (reanalysis distribution), and on the calibration of the models not being improved by the model developers.We see no way of grounding such assumptions given the available data.

Similar plots, without confidence intervals, also appear in recently published publications on this topic, including the quite influential "The rise of data-driven weather forecasting: A first statistical assessment of machine learning-based weather forecasts in an operational-like context" (Ben Bouallègue et al., 2024, DOI https://doi.org/10.1175/BAMS-D-23-0162.), developed by the ECMWF team. We would therefore suggest maintaining Figures 7-10 as they are in this respect.

Figure 8: Ditto both comments from Fig 7

See our two replies above.

Line 261-277: While this is interesting, it feels a little out of place since you haven't discussed FuXi anywhere else, and in the next section you once again stop referring to FuXi. I feel like you should either move these paras to the appendix so that the FuXi and other reanalysis initialized models are selfcontained in the appendix, or you should add some acknowledgement of their analysis to the opening sentences of the conclusion to make the transition from these paragraphs to the conclusion less jarring.

We agree with the reviewer that these reflections might belong better to other sections in the manuscript. We will therefore shorten them down to one paragraph, and move them to the discussion section, after an overview of the main results. We still think there is some value in maintaining this discussion point in the main body, though, as it might be of interest to a number of readers.

**Discussion and Conclusions**

Line 295: It also looks to me like Pangu does better at cold extremes, and for hot extremes GraphCast is better over the oceans while Pangu is better over land (based on Fig 6).

Thank you for pointing this out, we will add this information and a more in depth analysis of Figure 6 to our discussion.

Line 298-299: "the choice of best model depends strongly on region, lead time, type of extreme and in some cases even level of extremeness". I feel like this might not be quite the right way to put it – to me this implies that there is strong variability in the magnitude of the differences between the model's scores with region, lead, extreme type etc., but the magnitude of the differences between the models is not clear form most of your figures. All we know is that there is a lot of variability in which is best with region, lead time etc., but not by how much it is best. I suspect that you meant that with your wording, but I'm a bit worried it could be misinterpreted and suggest you revise this statement.

We agree that the wording here was not ideal, and that some of our conclusions might have been affected by the noise in the results. By implementing the significance test illustrated above, we have identified some clearer and more robust patterns. Thus, we now place greater emphasis on these robust results rather than on general statements such as the one above. We now highlight, in particular, how the quality of data-driven forecasts declines at longer lead times, likely due to blurring (Bonavita, 2024; Price, 2024). Moreover, we notice a clear trend towards better data-driven forecasts closer to the Equator, and worse at higher latitudes, which we link to the use of area-latitude weights in the loss function (e.g. Lam et al., 2023, Chen et al., 2023). Lastly, we find that IFS HRES displays the best calibration overall in terms of tail reliability, and that the extreme weather forecasts generated by data-driven models might benefit from some form of debiasing.

Line 301-303: ". Ideally, we envisage a hybrid use of physics- and data-driven models to forecast extremes, with physics-based models being supplemented by data-driven models for those areas where data-driven models have been shown to be superior in terms of tail performance." - are the sizes of the differences in performance enough to justify this approach? I'm taking an operational forecast perspective here, and it feels like the potential gains would have to be more than marginal to justify this.

We agree with the reviewer that this statement was not well-supported in our manuscript. As we have added tests of statistical significance to all our metrics (Figure 1-6) and magnitude figures to the new Appendix A, we believe there is now a stronger support for this statement, at least for some regions. Moreover, several hybrid models have been developed in the last months, including the popular Neural GCM (KochKov, 2024). In general, we do not see many hinders to at least simpler forms of hybrid usage of data-driven and physical modes (e.g. a simple weighted model average between the two), given that the global forecasts generated by the machine learning models require only a single GPU and very limited computational time. Indeed, several meteorological institutes, including ECMWF and NOAA, are already producing pre-operational forecasts both with physical and data-driven models, which could easily be coupled together.

**Appendix**

The same comments apply to the appendix figures as to the figures in the main text: • Some measure of statistical significance would be useful • For figures A5 and A6, some

accompanying figures showing the magnitude of the differences between the models would help give perspective on the significance of the spatial patterns in the figure. • For figures A4, A5 and A6, a different colour scheme would work better I think – one where each of the models is given a different colour (not just shades of the same colour), and a scheme which is not the same as the one used for showing the magnitude of the differences in scores (since this has a value judgement attached to it).

As suggested by the reviewer, we are going to implement the same changes to the main and the Appendix. Additionally, we will add some figures to the new Appendix D showing the magnitude of the differences in grid pointwise performance corresponding to Figure 5 and 6 in the main.

As an additional step to improving the clarity of our study, we will expand our discussion to address the limitations of our extreme metrics, emphasising that every metric has weaknesses, and that any attempts to make overarching comparisons between models should account for a range of different metrics simultaneously, as well as look at the performance of the forecasting models for the whole distribution of the variables, and not just at the tails. Specifically, QQ-plots and other reliability checks are key here, since they could easily expose attempts to hedge extreme metrics such as the tail RMSE. We will further add some figures related to this point to a new Appendix B.

**References**

Arellano, M. 'PRACTITIONERS' CORNER: Computing Robust Standard Errors for Within-Groups Estimators'. *Oxford Bulletin of Economics and Statistics* 49, no. 4 (1987): 431–34. https://doi.org/10.1111/j.1468-0084.1987.mp49004006.x.

Benjamini, Yoav, and Yosef Hochberg. 'Controlling the False Discovery Rate: A Practical and Powerful Approach to Multiple Testing'. *Journal of the Royal Statistical Society. Series B (Methodological)* 57, no. 1 (1995): 289–300.

Bi, Kaifeng, Lingxi Xie, Hengheng Zhang, Xin Chen, Xiaotao Gu, and Qi Tian. 'Accurate Medium-Range Global Weather Forecasting with 3D Neural Networks'. *Nature*, 5 July 2023, 1–6. https://doi.org/10.1038/s41586-023-06185-3.

Bonavita, Massimo. 'On Some Limitations of Current Machine Learning Weather Prediction Models'. *Geophysical Research Letters* 51, no. 12 (2024): e2023GL107377. https://doi.org/10.1029/2023GL107377.

Bouallègue, Zied Ben, Mariana C. A. Clare, Linus Magnusson, Estibaliz Gascón, Michael Maier-Gerber, Martin Janoušek, Mark Rodwell, et al. 'The Rise of Data-Driven Weather Forecasting: A First Statistical Assessment of Machine Learning-Based Weather Forecasts in an Operational-like Context'. *Bulletin of the American Meteorological Society* 1, no. aop (29 February 2024). https://doi.org/10.1175/BAMS-D-23-0162.1.

Cameron, A. Colin, and Douglas L. Miller. 'A Practitioner's Guide to Cluster-Robust Inference'. *Journal of Human Resources* 50, no. 2 (31 March 2015): 317–72. https://doi.org/10.3368/jhr.50.2.317.

Chen, Lei, Xiaohui Zhong, Feng Zhang, Yuan Cheng, Yinghui Xu, Yuan Qi, and Hao Li. 'FuXi: A Cascade Machine Learning Forecasting System for 15-Day Global Weather Forecast'. *Npj Climate and*

*Atmospheric Science* 6, no. 1 (16 November 2023): 1–11.
https://doi.org/10.1038/s41612-023-00512-1.

Kochkov, Dmitrii, Janni Yuval, Ian Langmore, Peter Norgaard, Jamie Smith, Griffin Mooers, Milan Klöwer, et al. 'Neural General Circulation Models for Weather and Climate'. arXiv, 7 March 2024. https://doi.org/10.48550/arXiv.2311.07222.

Lam, Remi, Alvaro Sanchez-Gonzalez, Matthew Willson, Peter Wirnsberger, Meire Fortunato, Ferran Alet, Suman Ravuri, et al. 'Learning Skillful Medium-Range Global Weather Forecasting'. *Science* 382, no. 6677 (22 December 2023): 1416–21. https://doi.org/10.1126/science.adi2336.

Pischke, Joern-Steffen, and Joshua Angrist. *Mostly Harmless Econometrics: An Empiricist's Companion*. Princeton, 2009.

Price, Ilan, Alvaro Sanchez-Gonzalez, Ferran Alet, Tom R. Andersson, Andrew El-Kadi, Dominic Masters, Timo Ewalds, et al. 'GenCast: Diffusion-Based Ensemble Forecasting for Medium-Range Weather'. arXiv, 1 May 2024. https://doi.org/10.48550/arXiv.2312.15796.

Rasp, Stephan, Stephan Hoyer, Alexander Merose, Ian Langmore, Peter Battaglia, Tyler Russell, Alvaro Sanchez-Gonzalez, et al. 'WeatherBench 2: A Benchmark for the Next Generation of Data-Driven Global Weather Models'. *Journal of Advances in Modeling Earth Systems* 16, no. 6 (2024): e2023MS004019. https://doi.org/10.1029/2023MS004019.

Wilks, D. S. '"The Stippling Shows Statistically Significant Grid Points": How Research Results Are Routinely Overstated and Overinterpreted, and What to Do about It', 1 December 2016. https://doi.org/10.1175/BAMS-D-15-00267.1.

Xu, Wanghan, Kang Chen, Tao Han, Hao Chen, Wanli Ouyang, and Lei Bai. 'ExtremeCast: Boosting Extreme Value Prediction for Global Weather Forecast'. arXiv, 2 February 2024. https://doi.org/10.48550/arXiv.2402.01295.

---

## Author Comment (AC2)

**Answer to Review #2**

Dear Reviewer,

Thank you for the time you spent reviewing our article and for your constructive feedback. We welcome your suggestion to expand our analysis to explore possible reasons for regional and variable-based differences between models. In the first draft of our manuscript, we limited ourselves to a descriptive analysis of the results due to concerns related to the limited sample size of our test data and the lack of statistical significance of the results. However, following valuable comments from the other reviewer, we have now expanded our analysis to also include robust significant tests of all our metrics, for all regions and individual grid points (Figure 1-6). We believe that these tests may contribute to strengthen the results of our analysis and also help to clarify which of the patterns we identified are supported by robust empirical evidence.

The testing approach we employ is based on clustered standard errors  (Liang and Zeger (1986), Arellano (1987), and Cameron and Miller (2015)), a classic econometric approach specifically designed for observations correlated in time and space. The intuition behind the use of clustered standard errors is that since many of our extreme data-points come from adjacent grid points and from events close to each other in time the effective number of degrees of freedom for our tests is much smaller than the total number of available paired forecast differences. Thus, we account for this by inflating the standard errors, by introducing a clustering parameter, which takes into account the clustered nature of our extremes in space and time. Below, we provide an example figure (updated Figure 2) to illustrate our changes. Black borders indicate here that the performance of the model at a given lead time is statistically significantly different from IFS HRES, at the 5% significance level.

[Figure]

Similarly, we performed significance testing with time-clustered standard errors also for grid-pointwise comparisons, exemplified below in a new figure to be added to the manuscript where we evaluate statistically significant differences between the IFS HRES and the best of the machine learning models at each grid point. Here, we also make use of global false discovering rates (Wilks 2016, Benjamini and Hochberg, 1995) to correct for multiple testing and ensure robust statistical inference. As illustrated by Wilks (2016), this approach is also robust for spatially correlated values. As in Figures 1-3, blue shades indicate that the machine learning model is better than IFS HRES, whereas red shades indicate that IFS HRES is better. Gray shades indicate a lack of statistically significant differences.

[Figure]

Additionally, we plan to add corresponding figures showing the magnitude of the differences between the models, see example figure below.

[Figure]

**Cold extremes** **Hot extremes** **Windspeed extremes**

% difference in RMSE vs IFS HRES

longitude

% difference in RMSE vs IFS HRES

Besides the above changes, we are planning to specifically address your concerns in a number of additional ways, by thoroughly revising and expanding several sections of our manuscript:

- Figures 1-3 are going to include about a page each of discussion of the results, where we explore possible explanations for regional and lead-time based differences in performance between the models. We identify, in particular, two key drivers for these differences, namely: the 1) the presence of increased blurring for data-driven models in relation to extreme weather forecasts, and 2) a meridional pattern in the quality of data-driven forecasts, with the best performance closest to the Equator, and the worst performance at high latitudes, in most cases. Besides figures 1-6 and the magnitude figure exemplified above, we also find evidence of this behaviour in the additional figure below (Figure R1), where we plot the relative difference in tail RMSE (5% most extreme events) between the ML models and IFS HRES (y-axis) vs latitude (x-axis) for 1-10 days ahead forecasts, as in previous figures. Despite the presence of some noise, we can notice some recurrent convexity in the performance of machine learning models, especially at shorter lead times, with clear spikes of poor performance close to the Poles. We believe that this behaviour may be ascribed to the use of area weighted loss functions, which place greater emphasis on errors closer to the Equator rather than to the Poles in order to maximise the performance in standard area-weighted performance metrics.

[Figure]

**Fig. R1:** *Relative difference in tail RMSE (y-axis) vs. latitude (x-axis) for cold (a), hot (b), and windy extremes (c). The data points are computed based on the (a) 5% lowest 2m temperatures, (b) 5% highest 2m temperatures, and (c) 5% highest 10m wind speeds, respectively. Forecasts are shown for X1) 1 day, X2) 3 days, X3) 5 days, X4) 7 days, and X5) 10 days. Negative values of the relative difference indicate better performance than IFS HRES, while positive values indicate worse performance than IFS HRES.*

- We are going to rewrite our description of Figures 5-6 to place greater focus on statistically significant results, and also explore differences in performance between different variables. We link these differences to the previously identified patterns, and also explore alternative explanations for newly identified regional patterns (e.g. lack of key input variables as a driver of subpar performance in continental areas among data-driven models).

- We are going to expand our description of Figures 7-10 to link the calibration results to the rest of the analysis, and provide a more in depth evaluation of the tail reliability of data-driven models in different regions. We now also provide several possible justifications for the results we obtain.
- Lastly, we are going to expand and partially rewrite the discussion and conclusion section to place greater emphasis on statistically significant results. Wherever possible, we also back up possible explanations for these results with findings from previous literature.
- Additionally, we will also expand our discussion to address the limitations of our extreme metrics, emphasising that every metric has weaknesses, and that any attempts to make overarching comparisons between models should account for a range of different metrics simultaneously, as well as look at the performance of the forecasting models for the whole distribution of the variables, and not just at the tails. Specifically, QQ-plots and other reliability checks are key here, since they could easily expose attempts to hedge extreme metrics such as the tail RMSE. We will further add some figures related to this point to a new Appendix B.

In summary, we believe our results may be ascribed to a number of different causes, which we are also going to discuss in our manuscript:

1. The overall worsened performance of data-driven models for extremes compared to standard metrics of average performance is likely linked to the choice of loss functions (Xu et al., 2024, Olivetti and Messori, 2024) and globally smoothed multitask approaches which are explicitly designed to optimise those metrics rather than extreme metrics.
2. The relative decline in performance of data-driven models at longer lead times is tied to the phenomenon of blurring (Bonavita, 2024; Price et al., 2024), which appears to be more prominent for extremes than for the overall distribution of the variables.
3. Other possible explanations for the decreased performance at longer lead times include the use of multi-step approaches in training  (e.g. Bi et al., 2023), which may lead to compound errors in initial atmospheric states over time (Bonavita 2024), and the use of only the most recent time steps as inputs for the models.
4. The observed regional pattern of better performance in the Tropics and worse performance at higher latitudes is likely connected to the use of latitude-based area weights (e.g. Lam et al., 2023, Chen et al., 2023), which optimise performance closer to the Equator at the expense of performance at higher latitudes.
5. The weaker performance of data-driven models for windy extremes, and for temperature extremes in some specific regions, may be tied to the lack of key input variables such as snow coverage, precipitation and soil moisture.
6. The overall weaker performance of data-driven models for wind extremes compared to temperature extremes may be related to the separate training of 10m u-and v-wind, whose errors may be magnified when looking specifically at 10m windspeed.

**References**

Arellano, M. 'PRACTITIONERS' CORNER: Computing Robust Standard Errors for Within-Groups Estimators'. *Oxford Bulletin of Economics and Statistics* 49, no. 4 (1987): 431–34. https://doi.org/10.1111/j.1468-0084.1987.mp49004006.x.

Benjamini, Yoav, and Yosef Hochberg. 'Controlling the False Discovery Rate: A Practical and Powerful Approach to Multiple Testing'. *Journal of the Royal Statistical Society. Series B (Methodological)* 57, no. 1 (1995): 289–300.

Bi, Kaifeng, Lingxi Xie, Hengheng Zhang, Xin Chen, Xiaotao Gu, and Qi Tian. 'Accurate Medium-Range Global Weather Forecasting with 3D Neural Networks'. *Nature*, 5 July 2023, 1–6. https://doi.org/10.1038/s41586-023-06185-3.

Bonavita, Massimo. 'On Some Limitations of Current Machine Learning Weather Prediction Models'. *Geophysical Research Letters* 51, no. 12 (2024): e2023GL107377. https://doi.org/10.1029/2023GL107377.

Bouallègue, Zied Ben, Mariana C. A. Clare, Linus Magnusson, Estibaliz Gascón, Michael Maier-Gerber, Martin Janoušek, Mark Rodwell, et al. 'The Rise of Data-Driven Weather Forecasting: A First Statistical Assessment of Machine Learning-Based Weather Forecasts in an Operational-like Context'. *Bulletin of the American Meteorological Society* 1, no. aop (29 February 2024). https://doi.org/10.1175/BAMS-D-23-0162.1.

Cameron, A. Colin, and Douglas L. Miller. 'A Practitioner's Guide to Cluster-Robust Inference'. *Journal of Human Resources* 50, no. 2 (31 March 2015): 317–72. https://doi.org/10.3368/jhr.50.2.317.

Chen, Lei, Xiaohui Zhong, Feng Zhang, Yuan Cheng, Yinghui Xu, Yuan Qi, and Hao Li. 'FuXi: A Cascade Machine Learning Forecasting System for 15-Day Global Weather Forecast'. *Npj Climate and Atmospheric Science* 6, no. 1 (16 November 2023): 1–11. https://doi.org/10.1038/s41612-023-00512-1.

Kochkov, Dmitrii, Janni Yuval, Ian Langmore, Peter Norgaard, Jamie Smith, Griffin Mooers, Milan Klöwer, et al. 'Neural General Circulation Models for Weather and Climate'. arXiv, 7 March 2024. https://doi.org/10.48550/arXiv.2311.07222.

Lam, Remi, Alvaro Sanchez-Gonzalez, Matthew Willson, Peter Wirnsberger, Meire Fortunato, Ferran Alet, Suman Ravuri, et al. 'Learning Skillful Medium-Range Global Weather Forecasting'. *Science* 382, no. 6677 (22 December 2023): 1416–21. https://doi.org/10.1126/science.adi2336.

Olivetti, Leonardo, and Gabriele Messori. 'Advances and Prospects of Deep Learning for Medium-Range Extreme Weather Forecasting'. *Geoscientific Model Development* 17, no. 6 (21 March 2024): 2347–58. https://doi.org/10.5194/gmd-17-2347-2024.

Price, Ilan, Alvaro Sanchez-Gonzalez, Ferran Alet, Tom R. Andersson, Andrew El-Kadi, Dominic Masters, Timo Ewalds, et al. 'GenCast: Diffusion-Based Ensemble Forecasting for Medium-Range Weather'. arXiv, 1 May 2024. https://doi.org/10.48550/arXiv.2312.15796.

Rasp, Stephan, Stephan Hoyer, Alexander Merose, Ian Langmore, Peter Battaglia, Tyler Russell, Alvaro Sanchez-Gonzalez, et al. 'WeatherBench 2: A Benchmark for the Next Generation of Data-Driven Global Weather Models'. *Journal of Advances in Modeling Earth Systems* 16, no. 6 (2024): e2023MS004019. https://doi.org/10.1029/2023MS004019.

Wilks, D. S. '"The Stippling Shows Statistically Significant Grid Points": How Research Results Are Routinely Overstated and Overinterpreted, and What to Do about It', 1 December 2016. https://doi.org/10.1175/BAMS-D-15-00267.1.

Xu, Wanghan, Kang Chen, Tao Han, Hao Chen, Wanli Ouyang, and Lei Bai. 'ExtremeCast: Boosting Extreme Value Prediction for Global Weather Forecast'. arXiv, 2 February 2024. https://doi.org/10.48550/arXiv.2402.01295.